# A corrosion-resistant RuMoNi catalyst for efficient and long-lasting seawater oxidation and anion exchange membrane electrolyzer

Xin Kang[1], Fengning Yang[1,2], Zhiyuan Zhang[1], Heming Liu[1], Shiyu Ge[1], Shuqi Hu[1], Shaohai Li [1], Yuting Luo[1,3], Qiangmin Yu [1] ✉, Zhibo Liu[4], Qiang Wang[4], Wencai Ren [4], Chenghua Sun [5], Hui-Ming Cheng [4,6,7] & Bilu Liu [1] ✉

Direct seawater electrolysis is promising for sustainable hydrogen gas ($H_2$) production. However, the chloride ions in seawater lead to side reactions and corrosion, which result in a low efficiency and poor stability of the electrocatalyst and hinder the use of seawater electrolysis technology. Here we report a corrosion-resistant RuMoNi electrocatalyst, in which the in situ-formed molybdate ions on its surface repel chloride ions. The electrocatalyst works stably for over 3000 h at a high current density of 500 mA cm$^{-2}$ in alkaline seawater electrolytes. Using the RuMoNi catalyst in an anion exchange membrane electrolyzer, we report an energy conversion efficiency of 77.9% and a current density of 1000 mA cm$^{-2}$ at 1.72 V. The calculated price per gallon of gasoline equivalent (GGE) of the $H_2$ produced is $ 0.85, which is lower than the 2026 technical target of $ 2.0/GGE set by the United Stated Department of Energy, thus, suggesting practicability of the technology.

Nowadays, the overuse of fossil fuels has caused a serious energy and environmental crisis. Hydrogen ($H_2$) as one of the most promising energy carriers for a sustainable society has advantages in the conversion and storage of renewable energy, especially when generated through water electrolysis powered by renewable energy[1,2]. However, the worldwide utilization of $H_2$ requires efficient electrolysis of purified freshwater which accounts for <1% of the world's total water resources[3]. On the other hand, seawater is one of the most abundant natural resources on our planet and accounts for 96.5% of the water on the earth[4]. Seawater electrolysis has a plentiful supply of water and is compatible with offshore wind parks or photovoltaic plants[5,6]. Seawater electrolysis leads to several promising research directions, such as desalinated, direct, and alkalized or acidified seawater electrolysis. It has been a heavily studied topic about the economic value of seawater electrolysis. For example, some analyses suggest that direct seawater electrolysis is not economic favorable[4,7,8], while some other studies suggest that direct seawater electrolysis shows low cost with economic benefit[9,10]. At this stage, the community has not reached a convincing or fixed conclusion about the preferable method, and more studies are needed on this topic, especially on the development of high-efficient and durable electrocatalysts for seawater electrolysis at high current density (>200 mA cm$^{-2}$)[11]. Recently, Guo et al. designed a flow-type electrolyzer using abundant seawater resources[12]. In another work, Xie et al. reported the one-step hydrogen production from seawater[13], indicating that the direct use of seawater in an industrial water electrolysis system, especially the anion exchange membrane (AEM) electrolyzer, is desirable[9,14,15].

[1]Shenzhen Geim Graphene Center, Tsinghua-Berkeley Shenzhen Institute & Institute of Materials Research, Tsinghua Shenzhen International Graduate School, Tsinghua University, Shenzhen 518055, P.R. China. [2]Department of Physics, University of Oxford, Clarendon Laboratory, Parks Road, Oxford OX1 3PU, UK. [3]Department of Electrical and Computer Engineering, University of Toronto, 35 St George Street, Toronto, Ontario M5S 1A4, Canada. [4]Shenyang National Laboratory for Materials Science, Institute of Metal Research, Chinese Academy of Sciences, Shenyang 110016, P.R. China. [5]Department of Chemistry and Biotechnology, Swinburne University of Technology, Hawthorn, Hawthorn, VIC 3122, Australia. [6]Faculty of Materials Science and Engineering, Institute of Technology for Carbon Neutrality, Shenzhen Institute of Advanced Technology, Chinese Academy of Sciences, Shenzhen 518055, P.R. China. [7]Advanced Technology Institute, University of Surrey, Guildford GU2 7XH, UK. ✉e-mail: yu.qiangmin@sz.tsinghua.edu.cn; bilu.liu@sz.tsinghua.edu.cn

However, plenty of chloride ions ($Cl^-$) in seawater deteriorate the performance of electrocatalysts for the seawater electrolysis, especially at high current densities due to the following reasons[13]. First, chlorine evolution reaction (ClER) is competitive to oxygen evolution reaction (OER) at anode that lowers the OER selectivity and forms toxic chlorine[16]. Second, the strong binding energy between $Cl^-$ and active sites of the electrocatalysts accelerates catalyst corrosion and leads to poor durability[9,17,18]. Electrolysis at high current densities is crucial for practical applications[19], but the above problems become more serious than that at low current densities (<200 mA cm$^{-2}$)[11,20,21]. As a result, the highest current densities delivered constantly by most of the seawater electrocatalysts reported so far remain below the industrial requirements of 500 mA cm$^{-2}$ [22,23], and it is rare that the electrocatalysts work stably for over 200 h[24,25]. Therefore, it is critical to develop a corrosion-resistant electrocatalyst to prevent $Cl^-$ corrosion in the seawater electrolytes[26–28]. Some success has been made recently in this field. For example, Kuang et al.[29] have reported that the intercalation of sulfate and carbonate ions in a nickel-iron hydroxide electrocatalyst improves its corrosion resistance to $Cl^-$. Beside durability, how to increase the selectivity of anode over OER is another critical issue in seawater electrolysis. In this regard, chloride barriers are widely used, and some electrode selectivity to $O_2$ production was enhanced to ~100%[23,30,31]. For example, $SiO_2$ overlayer has been introduced as an effective barrier which blocks the transport of $Cl^-$ and increases the selectivity to the desired OER[22]. Ma et al. have studied the effect of a sulfate additive on stable alkaline seawater oxidation and found that sulfate anions are preferentially absorbed on the anode surface to repel $Cl^-$ and achieve high selectivity[32]. Although those strategies show selectivity improvement, the corrosion of the conductive substrate in the saline electrolyte is still challenging. For example, sulfate anions have been shown to accelerate the corrosion of the electrocatalyst substrate because metal sulfates are unstable products which would be further oxidized to hydroxides or chlorides, and sulfate anions are released again and restart another cycle, finally resulting in the degradation of the electrode[33,34]. These works inspire us to design catalysts with an anion that can repel $Cl^-$ but without accelerating electrode corrosion.

In this study, we report a highly-efficient and stable seawater electrocatalyst RuMoNi with in situ formed $MoO_4^{2-}$ on its surface which repels $Cl^-$ with no corrosive effect on the electrode. The $MoO_4^{2-}$ is formed by leaching Mo from the RuMoNi electrocatalyst during electrochemical reconstruction. $MoO_4^{2-}$ is stabilized for at least 3000 h (the time we measured) by the reversible dissolution and precipitation of $NiMoO_4$. This electrocatalyst has ~100% selectivity for OER in a 1.0 M KOH + seawater electrolyte. It operates stably for over 3000 h at 500 mA cm$^{-2}$ with a negligible decay rate of 0.64 μV h$^{-1}$, meaning that the cell voltage would suffer an increase as small as 56 mV after operation for 10 years. An AEM electrolyzer catalyzed by the RuMoNi electrocatalyst achieves seawater electrolysis of 1000 mA cm$^{-2}$ at 1.72 V with an energy conversion efficiency of 77.9%. The price per gallon of gasoline equivalent (GGE) of the $H_2$ produced is $ 0.85, which is lower than the target ($ 2.00 per GGE $H_2$) by 2026 from the U.S. Department of Energy (DOE).

## Results and discussion

### Preparation and characterization of the RuMoNi electrocatalyst

We used a two-step method including hydrothermal process and electrochemical activation to synthesize the RuMoNi electrocatalyst (Fig. 1a, see "Methods" and Supplementary Fig. 1 for details). After hydrothermal process, the product consists of an array of almost parallel nanorods whose surfaces are coated by uniform distributed nanoparticles (Supplementary Fig. 2). After the electrochemical oxidation process, the RuMoNi electrocatalyst maintains the nanorod morphology and has a porous surface (Fig. 1b, c and Supplementary Fig. 3). X-ray photoelectron spectroscopy (XPS) indicates that the electrocatalyst contains Ru, Mo, Ni, and O (Supplementary Figs. 4–7).

A high-resolution transmission electron microscopy (HRTEM) image (Fig. 1d) shows that the electrocatalyst is composed of Ni$_4$Mo (Fig. 1e), $RuO_2$ (Fig. 1f, g), amorphous reconstructed phases (Fig. 1h), and $NiMoO_4$ (Fig. 1i). From Fig. 1d, f, Ni$_4$Mo mainly takes up the inside region and serves as the substrate, where $RuO_2$ nanoparticles anchor on the outside surface. In addition, $RuO_2$ and the amorphous phase are under the coverage of $NiMoO_4$.

Ex situ XPS spectra of Ni 2$p$ show that the Ni atoms on the surface are oxidized from Ni$^0$ to Ni$^{2+}$ during electrochemical oxidation, namely the reconstruction process (Supplementary Fig. 7). Meanwhile, the Ni atoms inside nanorod keep at metallic state as Ni$_4$Mo phase during the process (Supplementary Fig. 8). We use X-ray absorption fine structure (XAFS) to study Ni valance transformation during the electrochemical oxidation. The white line peak shifts to the higher energy in the Ni K-edge X-ray absorption near edge structure (Supplementary Fig. 9a). Peaks at 1.5 and 2.1 Å scattering on Ni K-edge Fourier transformed-extended X-ray absorption fine structure (FT-EXAFS) are assigned to the first coordination shell of Ni-O and the second coordination shell of Ni-Ni[35], respectively (Supplementary Fig. 9b). The Ni$^0$ is oxidized in the alkaline saline electrolyte and the Ni-O scattering holds steadily after this process. Furthermore, in situ Raman spectroscopy shows peaks of NiOOH at 477 cm$^{-1}$ and 558 cm$^{-1}$ (Supplementary Fig. 10)[36,37]. For the amorphous phase on the surface, the above results indicate that it contains NiOOH. The energy dispersive spectroscopy (EDS) result reveals the uniform distribution of Ni, Ru, and Mo elements over the nanorods (Fig. 1j). The inductively coupled plasma optical emission spectroscopy (ICP-OES) results indicate the low content of Ru elements of 0.15 wt%. After the OER process, the XPS spectrum of Ru 3$d$ reveals the valence state of Ru$^{4+}$ 3$d_{5/2}$ and Ru$^{4+}$ 3$d_{3/2}$ at 280.10 eV and 284.40 eV, respectively (Supplementary Fig. 11), which is reported to be resistant to corrosion and oxidation in harsh environments with improved OER activity[38,39]. Accordingly, we conclude that the RuMoNi electrocatalyst consists of the $RuO_2$/NiOOH active phase, $NiMoO_4$ corrosion-resistant layer, and conductive Ni$_4$Mo substrate.

### OER performance in alkaline seawater at high current densities

We then investigated the OER performance of the RuMoNi electrocatalyst in a three-electrode cell. Figure 2a shows the catalytic activity of RuMoNi, commercial $RuO_2$, and Ni foam in a 1.0 M KOH + seawater electrolyte (Supplementary Fig. 12, Supplementary Table 1). The RuMoNi electrocatalyst only requires an overpotential of 245 mV to achieve a current density of 10 mA cm$^{-2}$, while overpotentials of $RuO_2$ and Ni foam are 362 mV and 422 mV (Fig. 2a). At 1.70 V vs. RHE, RuMoNi delivers a high current density of 1000 mA cm$^{-2}$, ~10 times higher than $RuO_2$ (108 mA cm$^{-2}$) and ~50 times higher than Ni foam (19 mA cm$^{-2}$). The specific OER current densities of the RuMoNi catalyst normalized by the electrochemical surface area (ECSA) (Supplementary Figs. 13, 14) show that RuMoNi has better intrinsic OER activity than those of the other two samples (Supplementary Table 2). The OER activities of RuMoNi were also studied in other electrolytes including 1.0 M KOH and 1.0 M KOH + 0.5 M NaCl. Figure 2b shows that the overpotentials of RuMoNi to achieve 100, 200, 500, and 1000 mA cm$^{-2}$ only change within 10 mV in the three different electrolytes (Supplementary Table 3). Therefore, the RuMoNi electrocatalyst operates well in alkaline, saline, and seawater electrolytes, showing that its activity is not influenced by adding $Cl^-$ in electrolytes. As the ClER may compete with the OER in seawater electrolytes, the selectivity for OER is key in seawater electrolysis[40]. We measured the oxygen generation Faradic efficiency of RuMoNi by the drainage gas collection method (Supplementary Fig. 15) and the results show a nearly 100% selectivity for the OER in the seawater electrolyte. The high Faradaic efficiency of the OER is confirmed by gas chromatography (Fig. 2c). Note that the Faradic efficiency of the OER

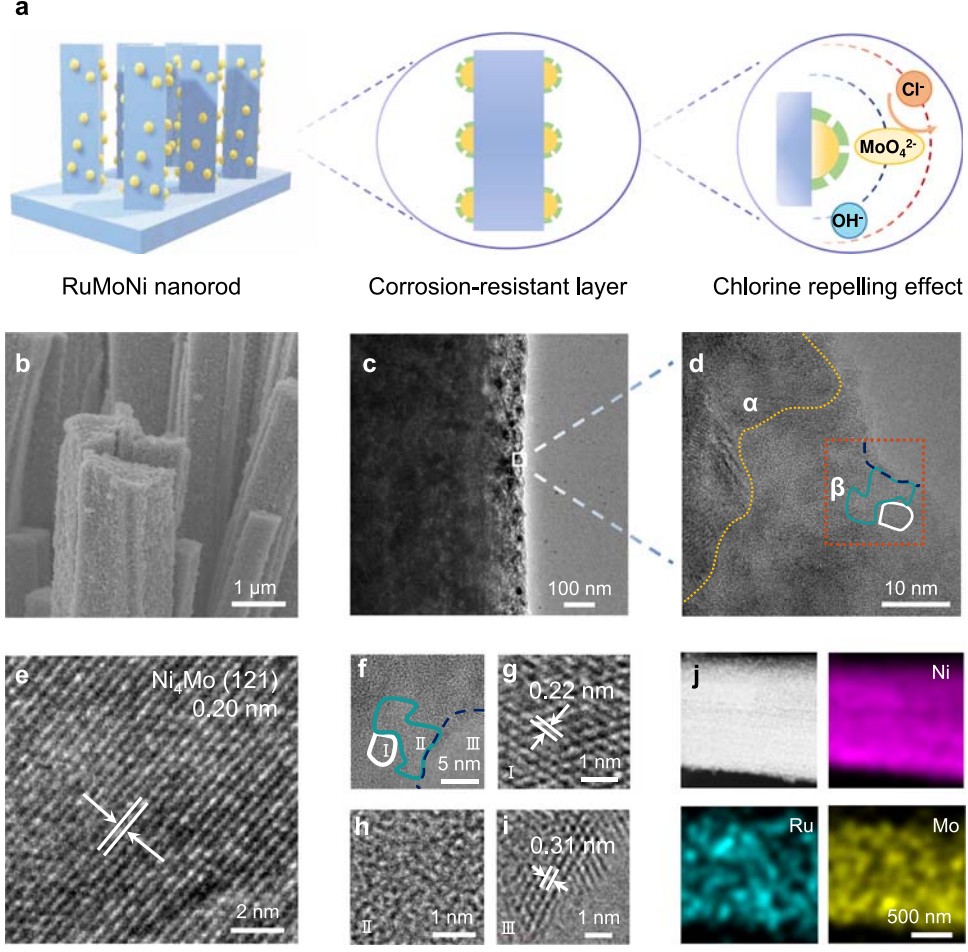

**Fig. 1 | Design principle and microscopic characterization of the RuMoNi electrocatalyst. a** A schematic showing the structure and corrosion-resistant strategy of the RuMoNi electrocatalyst. The light blue bar, yellow semicircle, and green dotted lines stand for nanorod-shape substrate, active sites, and corrosion-resistant layer, respectively. **b** SEM image of the as-prepared RuMoNi electrocatalyst. **c** TEM image of the RuMoNi nanorod. **d** HRTEM image of the RuMoNi electrocatalyst. The alpha area (α) corresponds to the region in (**e**), and the beta area (β) in the dashed red square corresponds to (**f**)–(**i**). **e** Lattice fringes of $Ni_4Mo$ (121) from the α region in (**d**). **f** The position of the three regions marked I, II, III from the β region in (**d**). **g** Lattice fringes of $RuO_2$ (111) corresponding to region I. **h** HRTEM image of the reconstructed surface corresponding to region II. **i** Lattice fringes of $NiMoO_4$ (220) corresponding to region III. **j** Energy dispersive X-ray spectroscopy maps showing the uniform distribution of Ni, Ru, and Mo elements.

remains unchanged after a 50 h-stability test, which shows the outstanding selectivity and stability of RuMoNi.

The electrochemical impedance spectroscopy (EIS) results demonstrate the efficient charge transfer of the RuMoNi electrocatalyst in 1.0 M KOH + seawater electrolyte (Fig. 2d, Supplementary Fig. 16), whose charge transfer resistance is smaller than those of the benchmark $RuO_2$ and Ni Foam based on the equivalent circuit in Supplementary Fig. 17 and fitted data in Supplementary Table 4[41]. In addition, the Tafel slope of RuMoNi (41.2 mV dec$^{-1}$) is the smallest one among the three samples (Fig. 2e), showing its fast OER kinetics. We used the indicator $\Delta\eta/\Delta\log|j|$ ($R_{\eta/j}$) to evaluate the OER performance of the electrocatalysts over a wide current range (Fig. 2f)[20]. The $R_{\eta/j}$ of RuMoNi ranged from 31.9 to 263.5 mV dec$^{-1}$ which is lower than that of $RuO_2$ (87.1–457.2 mV dec$^{-1}$) at current densities from 1 to 1000 mA cm$^{-2}$. The low $R_{\eta/j}$ indicates the highly efficient charge and mass transfer, and therefore good kinetics of the RuMoNi electrocatalyst over a wide range of current densities. Taken together, these results confirm that the RuMoNi electrocatalyst works well at both low and high current densities.

**Durability of the RuMoNi electrocatalyst under harsh conditions**

Seawater is a strongly corrosive environment with ~0.5 M Cl$^-$ which will bind to metal sites, resulting in the blocking of active sites and a

degradation of activity[18]. The durability of an electrocatalyst is a major challenge for the industrialization of seawater electrolysis. We therefore performed a series of experiments at different temperatures and in various electrolytes to study the durability of the RuMoNi electrocatalyst. The results show that it maintains a high activity with negligible performance decay after a stability test for 3000 h at a current density of 500 mA cm$^{-2}$ in a 1.0 M KOH + seawater electrolyte (Fig. 3a, Supplementary Fig. 18). We use the criteria $D_V = \frac{\bar{V}_2 - \bar{V}_1}{t}$ to evaluate the durability, in which $\bar{V}_1$ and $\bar{V}_2$ are averages of voltages from the first and last 10% operation time (details in "Methods"). As shown in Fig. 3b, the $D_V$ of RuMoNi is only 0.64 µV h$^{-1}$, which is smaller than the target set by the United States Department of Energy (U.S. DOE) (1.0 µV h$^{-1}$)[14]. We also studied the durability of the electrocatalyst at high temperatures. Figure 3c shows that the activity of RuMoNi electrocatalyst at 500 mA cm$^{-2}$ remains steady at temperatures of 40, 60, and 80 °C. During practical electrolysis, salt may accumulate in the electrolyte when seawater is continuously fed to the system and water is consumed to produce $H_2$ and $O_2$. To this end, we performed a chronopotentiometry (CP) test in the electrolyte with a three times higher NaCl concentration than seawater, using a 1.0 M KOH + 2.0 M NaCl electrolyte, where the test at 500 mA cm$^{-2}$ for 300 h shows no decay (Fig. 3d). Linear sweep voltammetry (LSV) curves before and after the CP test show a negligible change and confirm the durability of the

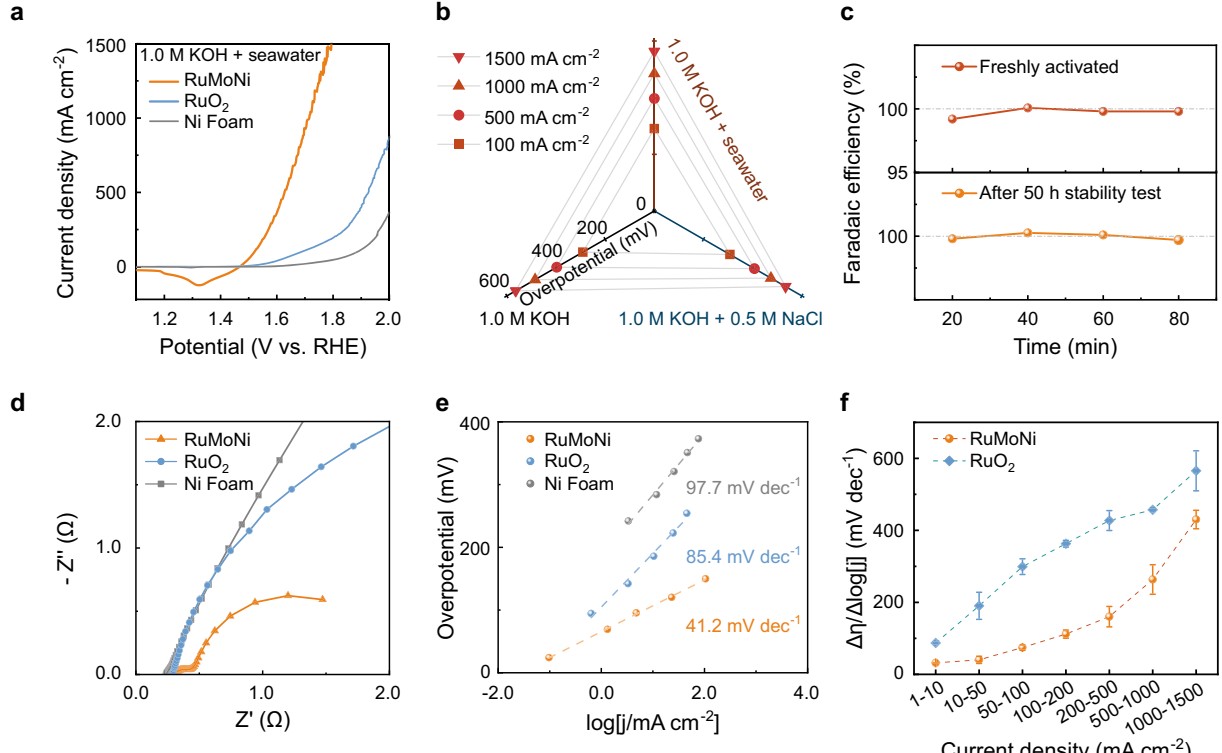

**Fig. 2 | OER performance in alkaline seawater at high current densities.**
**a** Polarization curves of RuMoNi, RuO$_2$, and Ni Foam in a 1.0 M KOH + seawater electrolyte at a scan rate of 5 mV s$^{-1}$ with 85% *iR* correction. The peak of RuMoNi near 1.3 V corresponds to the reduction peak of Ni sites. **b** The overpotentials of the RuMoNi electrocatalyst to achieve current densities of 100, 500, 1000, and 1500 mA cm$^{-2}$ in 1.0 M KOH + seawater, 1.0 M KOH + NaCl, and 1.0 M KOH electrolytes. **c** Oxygen generation Faradaic efficiency of the RuMoNi electrocatalyst in

1.0 M KOH + seawater electrolyte at 200 mA cm$^{-2}$. **d** Electrochemical impedance spectra of RuMoNi, RuO$_2$, and Ni Foam with raw impedance data reported as symbols and fitted data as lines. Enlarged spectra are shown in Supplementary Fig. 16. **e** Tafel slopes of the three electrocatalysts with chronoamperometry measurements. **f** The R$_{\eta/j}$ of the RuMoNi and RuO$_2$ electrocatalysts in different current density ranges.

RuMoNi (Supplementary Fig. 19). By testing the corrosion rate of electrodes and changes of open circuit potential (OCP) along the CP test, the RuMoNi with high corrosion resistance shows a negligible degradation over time (Supplementary Table 5, Supplementary Fig. 20). Meanwhile, the nanorod structure of the RuMoNi is maintained after long-term electrolysis, indicating its structural robustness (Supplementary Fig. 21). We also studied the selectivity of the catalyst in different conditions using UV-vis spectroscopy, and the absence of an iodine peak caused by hypochlorite as a ClER product suggests the good OER selectivity of the catalyst during the aforementioned stability test (Supplementary Figs. 22, 23). As a result, the RuMoNi electrocatalyst shows good stability without corrosion, selectivity deficiency, or activity degradation in highly saline electrolytes over 3000 h.

To assess the life of the RuMoNi electrocatalyst, the current density and duration of the durability test were compared with state-of-the-art alkaline seawater OER electrocatalysts (Fig. 3e, Supplementary Table 6). The RuMoNi electrocatalyst with a current density of 500 mA cm$^{-2}$ and test duration of 3000 h outperforms all other electrocatalysts and therefore sets up a higher bar for seawater electrolysis. To eliminate the influence of different test current densities and times on evaluating the durability of the electrocatalysts, we propose a new criterion, the degradation rate of activity ($D_A$), which is calculated by $D_A = \frac{D_V}{V_1} = \frac{V_2 - V_1}{V_1 t}$. Based on the result from the CP test, $D_V$ and $D_A$ of RuMoNi are 0.64 µV h$^{-1}$ and 3.98 × 10$^{-7}$ h$^{-1}$. It is speculated that after 10-year operation, the voltage would increase by only ~56 mV (Supplementary Fig. 24), which suggests the stability of RuMoNi to a certain extent. Note that the $D_V$ and $D_A$ values of our sample are an order of magnitude lower than those of other reported electrocatalysts in alkaline seawater electrolytes (Fig. 3f, Supplementary Table 7).

Therefore, it would be possible to use RuMoNi for industrial seawater electrolysis in an AEM electrolyzer which is suitable for operation in impure water[14]. In addition, the parameters $D_V$ and $D_A$ are meaningful for evaluating the electrocatalysts in different industrial operations.

## The mechanism of durability and selectivity of the RuMoNi electrocatalyst

The RuMoNi electrode is hydrophilic, while the liquid contact angle of Ni foam reaches 116°, indicating the good electrolyte wettability of RuMoNi (Fig. 4a). Based on the solid-liquid-gas interface theory, roughness at the micro- and nanoscales would reduce the contact area between a bubble and the electrode and hence decrease the catalyst-bubble adhesion force, resulting in a good mechanical stability of the electrocatalyst[42,43]. Fast bubble detachment will increase mass transfer and facilitate the reaction kinetics, especially at a high current density[28,44]. Tafel plots show the corrosion potential of RuMoNi is more positive than that of Ni foam in a 1.0 M KOH + seawater electrolyte (Fig. 4b), indicating a lower corrosion tendency and the improved corrosion resistance of the RuMoNi electrocatalyst compared to a bare Ni substrate. The ICP-OES results show that the atomic concentrations of Ru and Mo in the electrolyte remain constant during the 150 h CP test at a current density of 500 mA cm$^{-2}$ (Fig. 4c), suggesting its corrosion resistance at high current densities. In the Mo 3$d$ XPS spectrum (Fig. 4d), a doublet is observed at 230.4 and 233.5 eV (Mo$^{6+}$ 3$d_{5/2}$ and Mo$^{6+}$ 3$d_{3/2}$) which originates from NiMoO$_4$[40] and is consistent with the XRD result of NiMoO$_4$ (PDF # 33-0948) (Fig. 4e). The XPS and XRD results show the existence of corrosion-resistant MoO$_4^{2-}$ on the electrode surface.

By probing the concentration of MoO$_4^{2-}$ absorbed on the electrode surface, we find that MoO$_4^{2-}$ tends to accumulate near the

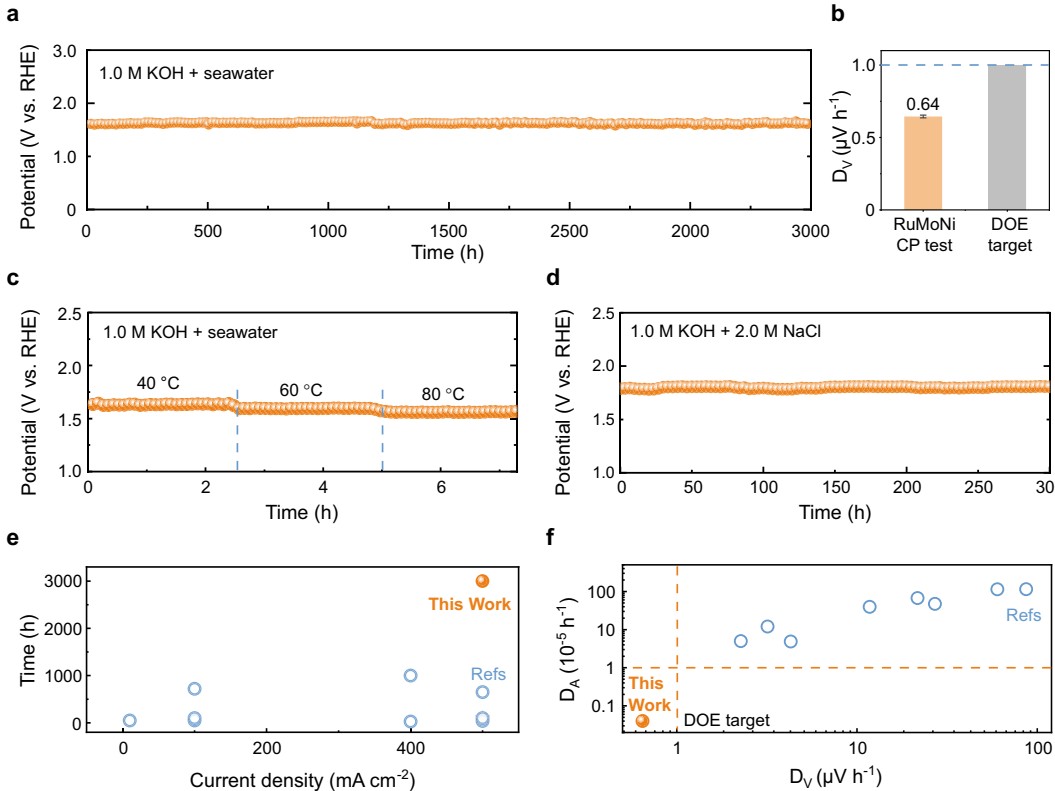

**Fig. 3 | Durability tests of the RuMoNi electrocatalyst in different practical conditions. a** Durability test for the OER for over 3000 h recorded at a current density of 500 mA cm$^{-2}$ in a 1.0 M KOH + seawater electrolyte. **b** Comparison of $D_V$ of RuMoNi to the U. S. DOE target. The error bar represents the standard deviation of the potentials. **c** Durability test for the OER at 40, 60, and 80 °C in a 1.0 M KOH + seawater electrolyte at a current density of 500 mA cm$^{-2}$. **d** Durability test for the OER in a 1.0 M KOH + 2.0 M NaCl aqueous solution at a current density of 500 mA cm$^{-2}$ for 300 h. **e** A comparison of the reported highest current density and the longest test duration of the durability tests achieved by other electrocatalysts and this work. **f** The $D_V$ and $D_A$ values calculated from the data of durability tests of other electrocatalysts and this work (All the data are from alkaline seawater electrolysis and details are shown in Supplementary Tables 6, 7).

surface and this phenomenon is more evident after applying an anode voltage (Fig. 4f)[45,46]. Combined with the constant atomic concentration of Mo in the electrolyte detected by ICP-OES, the reversible dissolution and precipitation of MoO$_4^{2-}$ and NiMoO$_4$ at the electrochemical interface is responsible for stabilizing the MoO$_4^{2-}$. Under the electric field during electrolysis, the multivalent MoO$_4^{2-}$ anions are preferentially absorbed on the anode through electrostatic force, and the enriched MoO$_4^{2-}$ anions near the anode surface repels and blocks Cl$^-$ by electrostatic repulsion[47]. Thus, the MoO$_4^{2-}$ pushes the Cl$^-$ away from the anode surface and increases the corrosion resistance of electrocatalyst[29]. Meanwhile, the strong hydrogen bonding between OH$^-$ and Ni$_4$Mo surface prevents electrostatic repulsion from impeding the OH$^-$ attack and maintain fast OER kinetics[32]. Because MoO$_4^{2-}$ by itself is not an oxidizing agent in a basic or neutral solution, it cannot polarize to corrode the electrode substrate[48]. MoO$_4^{2-}$ has also been demonstrated to give a protective effect in a Cl$^-$ containing solution by reducing Cl$^-$ adsorption and penetration of the corrosion-resistant layer[49,50], which was evidenced by the durability of NiMo catalyst at a current density of 500 mA cm$^{-2}$ for more than 250 h in the CP test (Supplementary Fig. 25). Therefore, the electrocatalyst with an absorbed MoO$_4^{2-}$ layer repels Cl$^-$ from catalyst surface and inhibits corrosion of the substrate, which results in the high OER selectivity and long life of the RuMoNi in seawater electrolysis.

## Performance of an AEM seawater electrolyzer catalyzed by RuMoNi catalysts

We further assembled an alkaline seawater AEM electrolyzer catalyzed by RuMoNi||RuMoNi as shown in Fig. 5a, b, and Supplementary Fig. 26. The AEM electrolyzer needs a cell voltage of only 1.72 V to reach a current density of 1.0 A cm$^{-2}$, which is 5 times higher than that of an electrolyzer using commercial RuO$_2$||Pt/C at the same cell voltage (Fig. 5c). Consequently, the performance of the AEM electrolyzer catalyzed by RuMoNi is improved compared to those catalyzed by commercial electrocatalysts. To evaluate the durability of the electrolyzer assembled with RuMoNi, we find that it operates at 500 mA cm$^{-2}$ for over 240 h, during which there is a negligible increase of voltage (Fig. 5d) and the catalyst retains its high selectivity during electrolysis (Supplementary Fig. 27). To give a comprehensive assessment of the alkaline seawater electrolyzer enabled by RuMoNi, we compare its performance with those of reported state-of-the-art catalysts in the literature[51]. As shown in Fig. 5e and Supplementary Table 8, the AEM seawater electrolyzer using the RuMoNi electrocatalyst has the best activity, highest H$_2$ production rate, longest durability test, and highest cell efficiency, standing out from the other reported AEM seawater electrolyzers. This improved performance confirms the possibility of industrialized AEM alkaline seawater electrolysis. We calculated the cell efficiency of RuMoNi|| RuMoNi, which shows a record 77.9% at a current density of 500 mA cm$^{-2}$ at 1.61 V, in comparison with 57.15% of RuO$_2$||Pt/C. As an important factor showing the economic efficiency of the electrolyzer, the price per GGE H$_2$ is as low as $ 0.85 according to the calculations in Supplementary Note 1, which is less than half of the target of $ 2.00 by 2026 from the U.S. DOE[52].

In summary, we have synthesized a RuMoNi electrocatalyst for highly-efficient, selective, and durable alkaline seawater electrolysis at a high current density. The in situ formed MoO$_4^{2-}$ absorbing on catalyst surface and repelling Cl$^-$, in addition to a corrosion-resistant layer consisting of NiMoO$_4$, is responsible for

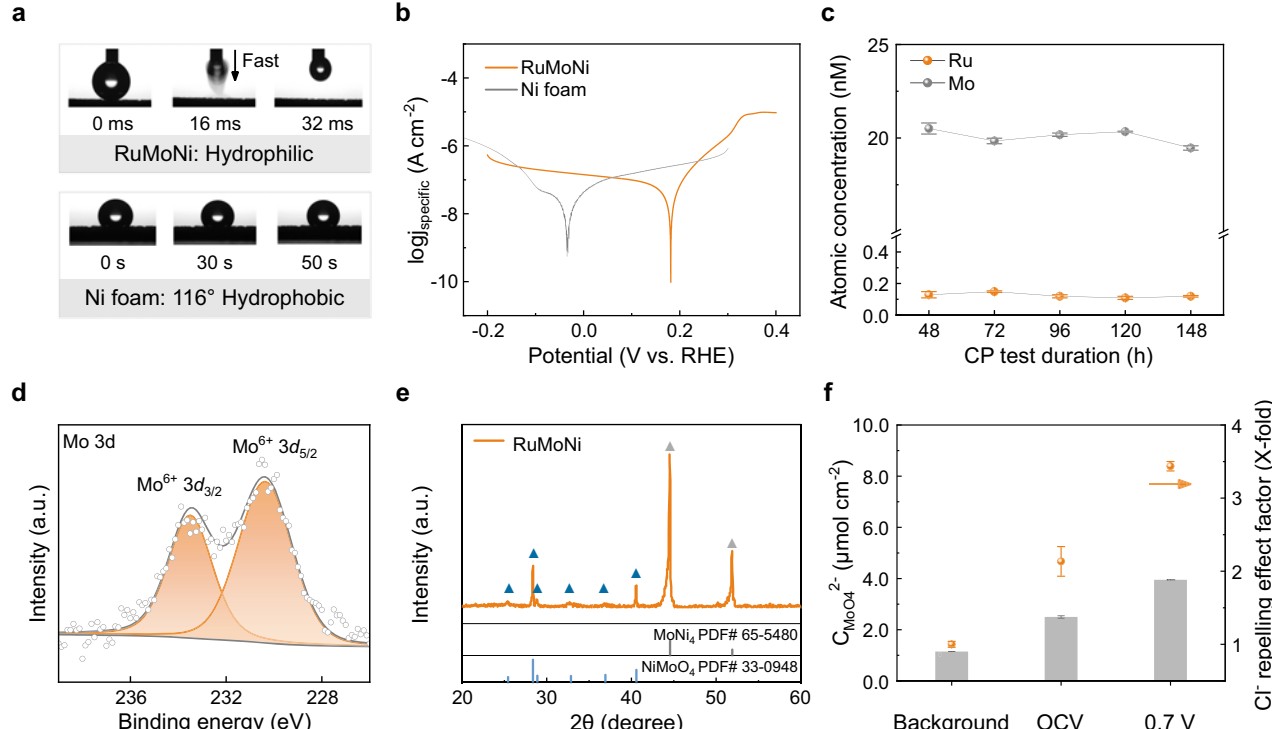

**Fig. 4 | Corrosion-resistant mechanism of the RuMoNi electrocatalyst. a** Photos showing the dynamic wettability on RuMoNi and Ni foam. The droplets are 1.0 M KOH + 0.5 M NaCl with an identical volume of 4 µL. **b** Tafel plots of RuMoNi and Ni Foam in a 1.0 M KOH + 0.5 M NaCl electrolyte. The $j_{specific}$ is the current density normalized by ECSA. **c** Atomic concentrations of Ru and Mo in the electrolyte during the chronopotentiometry test for 150 h. Each data is derived from at least three experiments, and error bars represent the standard deviation of three samples. **d** XPS spectrum of Mo 3d. The unit of the intensity is arbitrary units (a.u.). **e** XRD pattern of RuMoNi. **f** The adsorbed $MoO_4^{2-}$ concentration on carbon paper (background group) and RuMoNi (group of OCV and 0.7 V) normalized by electrode area and corresponding $Cl^-$ repelling effect factors. Carbon paper was used as the background group because of the plain interaction between carbon paper and $MoO_4^{2-}$ from the electrolyte without its dissolution and precipitation on the RuMoNi surface. Each solution was sampled three times, and error bars correspond to standard deviation of three samples.

the high OER selectivity and corrosion resistance to $Cl^-$ anions in seawater. The electrocatalyst sustains catalysis at 500 mA cm$^{-2}$ for over 3000 h with a negligible decay rate of 0.64 µV h$^{-1}$, which means the cell voltage would suffer a voltage increase as low as 56 mV over a 10-year-long operation. The seawater AEM electrolyzer assembled using RuMoNi achieves an improved performance of a high activity (1.72 V at 1000 mA cm$^{-2}$, industrial conditions), high cell efficiency (77.9% at 500 mA cm$^{-2}$), and long-life (500 mA cm$^{-2}$ over 240 h).

## Methods
### Chemicals
Nickel (II) nitrate hexahydrate ($Ni(NO_3)_2 \cdot 6H_2O$, AR, Guangdong Guanghua Sci-Tech Co., Ltd, China), ammonium molybdate tetrahydrate (($NH_4)_6Mo_7O_{24} \cdot 4H_2O$, AR, Shanghai Aladdin Biochemical Technology Co., Ltd, China), ruthenium chloride hydrate ($RuCl_3 \cdot xH_2O$, AR, Shanghai Aladdin Biochemical Technology Co., Ltd, China), platinum nominally 20% on carbon black (20% Pt/C, Alfa Aesar Chemical Co., Ltd), ruthenium oxide ($RuO_2$, 99.9% metals basis, Ru ≥ 84.5%, Shanghai Aladdin Biochemical Technology Co., Ltd.), Nafion (5%, D520, E. I. Dupont de Nemours and Company), sodium chloride (NaCl, AR, 99.5%, Shanghai Macklin Biochemical Co., Ltd), potassium hydroxide (KOH, GR, 95%, Shanghai Macklin Biochemical Co., Ltd) were used without further purification. Ni foam (0.5 mm thick, Linyi Gelon LIB Co., Ltd), Ti foil (0.2 mm thick, 99.99%, Zhongnuo Co., Ltd), Pt wire (1 mm diameter, 99.99%, Zhongnuo Co., Ltd) were used as received. Ultrapure Direct-Q water (18.2 MΩ cm$^{-1}$) was used to prepare all aqueous solutions and wash samples.

### Synthesis of the RuMoNi electrocatalyst
We synthesized the RuMoNi electrocatalyst by a two-step method. First, Ru-doped $NiMoO_4$ was synthesized on Ni foam through a hydrothermal process. By dissolving $Ni(NO_3)_2 \cdot 6H_2O$, $(NH_4)_6Mo_7O_{24} \cdot 4H_2O$, and $RuCl_3 \cdot xH_2O$ in 30 mL deionized water, we obtained a solution with 40 mM of Ni, 10 mM of Mo, and 0.5 mM of Ru in a hydrothermal autoclave. A piece of Ni foam (30 mm × 10 mm × 0.5 mm) was sonicated in a 1.0 M HCl aqueous solution for 40 min to remove the surface oxide layer and then washed with deionized water to remove residual HCl. After adding the cleaned Ni foam to the mixed solution, the system reacted for 6 h at 150 °C in an oven. Then, the Ru-doped $NiMoO_4$ on Ni foam was reduced in an $Ar/H_2$ atmosphere. The catalyst was put into a quartz tube furnace and purged with Ar (300 sccm) before annealing in an $H_2/Ar$ atmosphere (v/v, 5 sccm/95 sccm) at 500 °C for 30 min. After the annealing process, the $H_2$ was turned off and the furnace cooled to room temperature in an Ar flow. Second, the above synthesized precatalyst went through electrochemical activation by chronopotentiometry (CP) at a current density of 50 mA cm$^{-2}$ for 10 h in the electrolyte corresponding to the electrochemical test. According to XPS spectra of the RuMoNi precatalyst and electrocatalyst, during the reconstruction process, $Ru^0$ is oxidized to $Ru^{4+}$ (Supplementary Fig. 5, Supplementary Fig. 11). $Ni^0$ on the surface is oxidized to $Ni^{2+}$, and $Ni^0$ exists in the interior region (Supplementary Fig. 7b, Supplementary Fig. 8b). $Mo^0$, $Mo^{4+}$, and $Mo^{6+}$ exist in RuMoNi precatalyst, and $Mo^{6+}$ in RuMoNi catalyst (Supplementary Fig. 6, Fig. 4d). Results from HRTEM (Fig. 1d–i), XRD (Fig. 4e), and Raman (Supplementary Fig. 10) reflect the composition changes during the

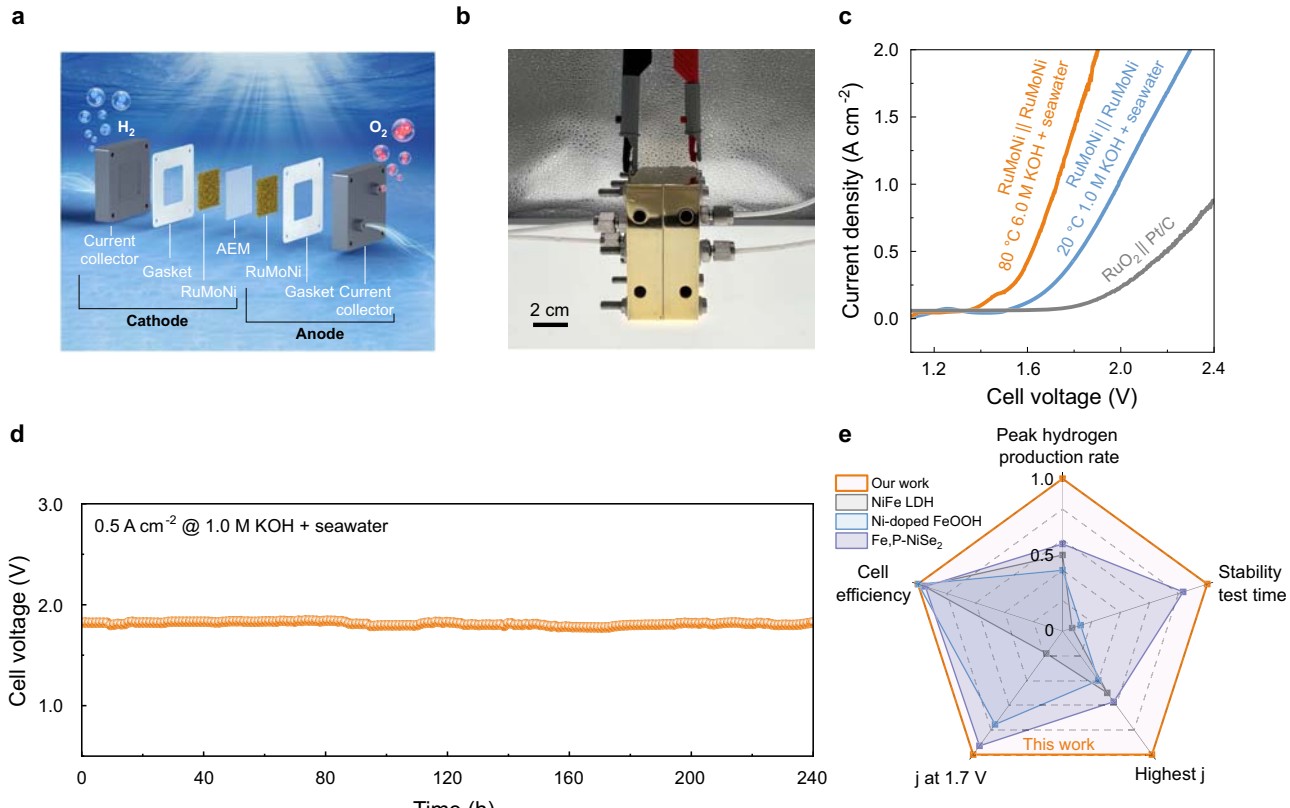

**Fig. 5 | AEM electrolyzer performance in alkaline seawater. a** Schematic of the AEM electrolyzer components and H₂ production by the AEM electrolyzer in alkaline seawater. **b** Photograph of the AEM electrolyzer. **c** Polarization curves normalized by the electrode area of RuMoNi || RuMoNi in 1.0 M KOH + seawater at 20 °C and 6.0 M KOH + seawater at 60 °C with RuO₂ || Pt/C as a comparison. **d** Durability test of the AEM electrolyzer using RuMoNi || RuMoNi at 500 mA cm⁻² in a 1.0 M KOH + seawater electrolyte. **e** Comprehensive comparison of the performance of the electrolyzer catalyzed by RuMoNi || RuMoNi with reported seawater AEM electrolyzers in a radar chart whose axes have different meanings and units (see Supplementary Table 8 for the references where the specific values of NiFe LDH, Ni-doped FeOOH, and Fe,P-NiSe₂ are reported).

reconstruction process. Ru on the surface reconstructs to RuO₂ active phases. Molybdenum oxides dissolve and form MoO₄²⁻ in the electrolyte. NiOOH and NiMoO₄ form on the surface. Ni₄Mo exists in the interior region. The NiMo electrocatalyst was synthesized by the same method as RuMoNi but without adding RuCl₃·xH₂O, and followed the same process of electrochemical activation.

## Materials characterization

The morphology of the samples was examined by SEM (5 kV, Hitachi SU8010, Japan). HRTEM analyses were carried out at an electron acceleration voltage of 300 kV (FEI Titan Cubed Themis G2 300, USA). Structural and chemical analyses of the samples were performed by powder XRD (Cu Kα radiation, λ = 0.15418 nm, Bruker D8 Advance, Germany) and XPS (monochromatic Al Kα X-rays, PHI5000VersaProbeII, Japan). The elemental composition was determined by ICP-OES (SPECTRO ARCOS II MV, Germany). The sample named OCV was immersed in an oxygen-saturated 1.0 M KOH + seawater electrolyte for 30 min.

## XAFS measurements

X-ray adsorption structure (XAS) spectra at the Ni K-edges were recorded at the BL11B beamline of the Shanghai Synchrotron Radiation Facility (SSRF). The beam current of the storage ring was 220 mA in a top-up mode. The incident photons were monochromatized by a Si (111) double-crystal monochromator, with an energy resolution ΔE/E ~1.4 × 10⁻⁴. The spot size at the sample was ~200 μm × 250 μm (H × V). The XAS spectra of the samples at Ni K-edges were calibrated by the Ni foil reference (8333 eV). The RuMoNi powders were exfoliated from the Ni foam and then loaded onto carbon paper. The electrochemical activation was conducted by CHI 660E electrochemical workstation. XAFS spectra at the Ni K-edge were collected in the fluorescence mode with a Lytle ionization chamber filled with Ar.

## In situ Raman measurements

In situ Raman spectra were collected using a 532 nm laser excitation with a beam size of ~1 μm (Horiba LabRAB Evolution, Japan). Measurements were performed with a gold-coated Ti foil that was loaded with catalyst powder as the working electrode using a homemade electrochemical cell setup. A Pt wire was used as the counter electrode and a Ag/AgCl electrode as the reference electrode in a 1.0 M KOH + seawater electrolyte. The scattered light was collected by a ×60 water immersion objective lens and then directed to a charge-coupled device (CCD) detector. Before the experiments, the monochromator was calibrated by the 520.7 cm⁻¹ peak of silicon.

## Electrochemical measurements

Electrochemical measurements were performed using a potentiostat (VMP-3, Biologic) with a three-electrode electrochemical cell. The electrochemical tests were carried out in a constant temperature laboratory at 20 ± 2 °C unless otherwise specified. After adding KOH into natural seawater and settling down the precipitate, the transparent electrolyte was directly used as an electrolyte to perform electrochemical tests. In the following electrochemical test in the 1.0 M KOH + seawater electrolyte, natural seawater without KOH or precipitation was added into the

electrolyte. The electrolyte in all tests was oxygen saturated 1.0 M KOH + seawater unless otherwise specified. A synthesized RuMoNi electrode (geometric area 1 cm × 1 cm) was used as the working electrode. A graphite rod was used as the counter electrode, and the Hg/HgO (1.0 M KOH) with a salt bridge was used as the reference electrode. All LSV data were from the backward scan of the cyclic voltammetry (CV) to eliminate the signal from the oxidation of Ni and compensated by an 85% IR correction with the distance between the working electrode and reference electrode fixed by a Luggin Capillary. The current densities were normalized by geometrical surface area. The pH values of 1.0 M KOH (14.13) and 1.0 M KOH + seawater (14.10) were tested by pH meter, and this value was also calculated to be 14.06 according to reference[53]. In this work, we took the pH values of 14.10 to perform the potential conversion. Potentials were converted to potential vs. RHE after being measured vs. Hg/HgO by $E_{RHE} = E_{Hg/HgO} + 0.925$ V. The scan rate was 5 mV s$^{-1}$ for the CV tests. After 50 CV cycles of activation, the OER electrocatalytic activity was studied. We measured the weight of RuMoNi contained in the electrode (1 cm × 1 cm) using ICP and a subtraction method. The mass loading of RuMoNi was 3.6 mg cm$^{-2}$. For comparison, commercial RuO$_2$ with a mass loading of ~4 mg cm$^{-2}$ was drop-cast onto Ni foam of the same size. Durability tests were conducted at a constant current density of 500 mA cm$^{-2}$ for 0–3000 h with a three-electrode electrochemical cell in a sequence of electrolytes at different temperatures by using electrolyte heater without climate chamber. During the durability test in 1.0 M KOH + seawater electrolyte, natural seawater without KOH or precipitation was pumped consistently into the electrolyte to simulate the real seawater electrolysis situation. During the durability test in 1.0 M KOH + 2.0 M NaCl, 2.0 M NaCl solution was pumped into the electrolyte. Tafel slope analyses were done with the steady-state response with a 100% IR drop compensation based on the overpotentials and current densities from 500 s chronoamperometry measurements at different applied voltages. The corrosion behaviors of different electrodes were studied in the three-electrode electrochemical cell using RuMoNi and Ni foam electrodes with a geometric area of 1 cm × 1 cm as the working electrodes. The polarization tests were performed in 1.0 M KOH + 0.5 M NaCl electrolyte. The open circuit potential (OCP) was measured after the electrode exposed in the electrolytes for several hours.

The double layer capacitance ($C_{dl}$) was calculated by CV measurements at scan rates from 10 to 50 mV s$^{-1}$ in the non-faradaic region in a 1.0 M KOH electrolyte, because of the negligible difference in $C_{dl}$ of RuMoNi electrode in 1.0 M KOH and 1.0 M KOH + seawater. The electrochemical active surface areas (ECSA) were obtained based on $C_{dl}$. The $C_{dl}$ is estimated using the equation: $C_{dl} = \frac{\Delta j}{\Delta \nu} = \frac{j_a - j_c}{2\nu}$, where $j_a$ and $j_c$ are the anodic and cathodic current densities, respectively, recorded at a potential of 1.125 V vs. RHE, and $\nu$ is the scan rate (Supplementary Fig. 11). An ideal planar electrode has a $C_{dl}$ of 0.04 mF cm$^{-2}$, defined as $C_s = 0.04$ mF cm$^{-2}$. The roughness factor ($R_f$) can be calculated using the equation: $R_f = \frac{C_{dl}}{C_s}$[54]. The specific current density ($j_{\text{specific}}$) was calculated by the following equation: $j_{\text{specific}} = \frac{j}{R_f}$.

## AEM electrolyzer fabrication

The AEM electrolyzer was assembled with an anode (2.0 cm$^2$), cathode (2.0 cm$^2$), and anion exchange membrane (AEM, X37-50 Grade T, Dioxide Materials). RuMoNi electrocatalyst with nickel foam was directly used as the monolith anode and cathode to construct the RuMoNi||RuMoNi AEM electrolyzer. For comparison, commercial RuO$_2$ powder with a polytetrafluoroethylene (PTFE) binder was coated on hot-pressed Ni foam and used as the anode. The loading mass of RuO$_2$ was approximately 4 mg cm$^{-2}$. The cathode was prepared from Pt/C with Nafion (5%, D520, E. I. Dupont de Nemours and Company) by the same method as the anode. The loaded amount of Pt was

~1 mg cm$^{-2}$. The assembly of the AEM electrolyzer required no additional process, such as heating or pressing. We studied the performance of the AEM electrolyzer in a 1.0 M KOH + seawater electrolyte at 20 ± 2 °C and 6.0 M KOH + seawater electrolyte at 60 °C ± 2 °C, using a potentiostat (Zahner XC, ZAHNER, Germany). The polarization curve was measured using LSV technology at a scan rate of 10 mV s$^{-1}$. The durability test was carried out by CP technology at a current density of 500 mA cm$^{-2}$ for over 240 h at 20 ± 2 °C, during which natural seawater without KOH or precipitation was directly added into the electrolyte. The temperature was controlled by electrolyte and electrolyzer heaters without climate chamber.

## Gas chromatography (GC) test and Faradaic efficiency measurements

We set up a gas-tight system consisting of an H-type electrolysis cell, GC (GC-MS, Thermo Fisher, USA), and tube system to evaluate the Faradaic efficiency of RuMoNi in a 1.0 M KOH + seawater electrolyte. The electrolysis was carried out at a constant current density of 500 mA cm$^{-2}$. Ar (99.999%) of 15 sccm was constantly purged into the anodic chamber which was connected to a gas-washing bottle to remove Cl$_2$ and vapor. We used a thermal conductivity detector to detect and quantify the generated oxygen.

## Definition of degradation rate of voltage ($D_V$) and degradation rate of activity ($D_A$)

We specified two criteria, degradation rate of voltage ($D_V$) and degradation rate of activity ($D_A$), which were calculated using the respective following equations, $D_V = \frac{\bar{V}_2 - \bar{V}_1}{t}$, $D_A = \frac{D_V}{\bar{V}_1} = \frac{\bar{V}_2 - \bar{V}_1}{\bar{V}_1 t}$. We took the average of the potential in the CP test or current density in a chronoamperometry (CA) test during the initial/final 10% time (defined as $\bar{V}_1/\bar{V}_2$, $\bar{j}_1/\bar{j}_2$, respectively) to denoise the fluctuations during long-term operation. Dividing the potential shift ($\bar{V}_2 - \bar{V}_1$) by the total time ($t$) normalized the results to the same time scale. Thereafter, $D_V$ indicates the increased rate of potential and stands for the total energy consumption during long-term electrolysis. The working condition of a high current density is harsher because of the rigorous bubble release, severe polarization, and high oxidation voltage, which decrease the durability and result in a high $D_V$. Taking account of the effect of a high current density, it is reasonable to divide the $D_V$ by the initial voltage (defined as $D_A$), which gives us the percentage change in the voltage during the durability test without being bothered by the different current densities. This can be used to estimate the degradation rate of activity. Additionally, we also define $D_A$ in a similar way to the CA test by replacing the potential shift ($\bar{V}_2 - \bar{V}_1$) with the current density shift ($\bar{j}_1 - \bar{j}_2$). Thereafter, $D_A$ can be defined as $\frac{\bar{j}_1 - \bar{j}_2}{\bar{j}_1 t}$.

## Reporting summary

Further information on research design is available in the Nature Portfolio Reporting Summary linked to this article.

# Data availability

The data generated in this study are provided in the Source data file. Source data are provided with this paper.

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

## Acknowledgements
We acknowledge financial support from the National Science Fund for Distinguished Young Scholars (No. 52125309), National Natural Science Foundation of China (No. 52188101), Guangdong Basic and Applied Basic Research Foundation (No. 2021A1515110829, No.2022B1515120004), the Guangdong Innovative and Entrepreneurial Research Team Program (No. 2017ZT07C341), and the Shenzhen Basic Research Project (No. JCYJ20200109144620815, No. WDZC20220812141108001). We also thank staff in BL11B beamline in Shanghai Synchrotron Radiation Facility (SSRF) for their technical assistance.

## Author contributions
X.K., Q.Y., and B.L. conceived the idea. X.K. and F.Y. synthesized the materials. X.K. performed most of material characterizations and electrochemical measurements. Z.Z. took part in the electrochemical measurements and discussions. Q.W., Z.L., and W.R. performed the TEM characterization. Q.Y., H.L., S.G., S.H., S.L., and Y.L. participated in the discussions and part of electrochemical tests. Q.Y. and B.L. supervised the project and directed the research. X.K., Q.Y., Z.L., W.R., C.S., H.C., and B.L. discussed and interpreted the results. X.K., Q.Y., and B.L. wrote the manuscript with feedback from other authors.

## Competing interests
The authors declare the following competing interests. B.L., X.K., Q.Y., and Z.Z. are inventors of a patent related to this work. Patents related to this research have been filed by Tsinghua Shenzhen International Graduate School, Tsinghua University. The University's policy is to share financial rewards from the exploitation of patents with the inventors. The remaining authors declare no competing interests.
