## [Peer Review File · Nature Communications]

REVIEWER COMMENTS

Reviewer #1 (Remarks to the Author):

Liu and co-workers report on a corrosion-resistant RuMoNi catalyst which is active for long-term seawater oxidation as well as seawater anion exchange membrane electrolyzer. First of all, seawater splitting especially when KOH is added, is tremendously more expensive than conventional water splitting, as such an electrolyzer requires a feed of KOH and seawater instead of only pure water, as the addition of only seawater will lead to increasing concentrations of all impurities. Therefore, the whole electrolyte must be replaced from time to time. If only pure water is used, the KOH can remain in the electrolyzer. Furthermore, seawater desalination (something that can always be done when seawater and electricity is available) adds basically no costs to the hydrogen production process. The authors have ignored all literature on this issue (see <https://doi.org/10.1039/D1EE00870F>, <https://doi.org/10.1039/D0EE03659E>, [10.1021/acssuschemeng.8b06779](https://doi.org/10.1021/acssuschemeng.8b06779),...). This does not mean that seawater splitting must not be performed academically, but the claims of the introduction are rather overestimated, and it is not solving any real-world problem making it not useful for society. This must be stated in the introduction and critical reports must be cited. Additionally, many other issues need to be clarified before this manuscript is considered for a high-impact journal such as Nature communication.

1. What is the oxidation state of Ru and Mo in RuMoNi precatalyst? Is it metallic? Also, the authors should clearly explain the bands related to the precatalysts and electrocatalysts in the Raman spectra.
2. In the main text, the authors state that Ru is doped in the precatalyst, which the reviewer presumes homogeneously distributed Ru in the structure (no related characterization is provided); however, in the electrocatalyst phase, RuO₂ can be found only at the surface. The reviewer wonders, if the only surface structure is reconstructed, what happens to the Ru that was beneath the top layer? Why can the authors only find Ni₄Mo in the inside region? It is also not clear what amorphous phase the authors find on the surface.
3. The construction process of RuNiMO should be described in detail. How (MoO₄)²⁻ are stabilized on the surface during the reaction.
4. Does the presence of nickel foam affect the electronic structure of the materials? Similarly, what is the Fe concentration in 1 M KOH? How much is Fe still present in the final electrocatalytic state? How does it influence electrocatalysis?

5. What is the pH of the 1 M KOH (+ seawater) solution? How was it measured? Which values were used to convert it to RHE? A recent couple of reports in ACS Energy Letters state that the pH meter is not a good method to evaluate precise pH values.

6. As the authors show that the used seawater contains a mixture of several inorganic substances (Mg^{2+} , Ca^{2+} , Na^{2+} , sulfate) as well as they may be organic mixtures and microorganisms. Such a complex mixture present in the seawater can deeply affect the reconstruction process and performance or can be adsorbed in the active phase. How did the author remove/or avoid such a phenomenon? The authors do not even seem to observe any adsorption or interference from the salts. If they were precipitated out before, then it would be an extra cost to separate these products. Did they consider the separation step in the price calculation? Again, this is, in fact, not a significant advantage in comparison to direct water splitting using purified water.

7. Tafel slopes must be measured via a steady-state method. How did the authors measure it? The description is missing in the manuscript.

8. Since $(\text{MoO}_4)^{2-}$ was the phase that was repelling Cl^- , did the authors use only RuMo or NiMo as references, especially NiMo, which would be more economically beneficial?

9. The reviewer missed the point of how the stability was performed. Was it a flow cell or did the authors add deionized water at a certain time interval during these tests?

10. The authors should also look at the recent reports, e.g., <https://doi.org/10.1039/C8EE00927A>, where only in an alkaline solution, a current density of 1 A can be generated in much lower potentials.

11. Depending on the 4-month stability period, the extrapolation of the data for 10 years seems to be mere speculation.

12. The C_{dl} was measured in KOH solution instead of KOH+seawater. Why? Does this influence the ECSA of the catalyst? The authors in the ESI state that “the ideal planar electrode has a C_{dl} of 0.04

mF cm^{-2} ” while they used a much more porous 3D nickel foam for the experiments.

Reviewer #2 (Remarks to the Author):

The idea is worthy of research but it needs to be done more detailed. I propose these suggestions:

1. There are lots of papers reported about the use of different chloride barrier layers to increase the selectivity of anodes over OER. However, in introduction section we do not see a concise bibliography of these works
2. In figure 2a, the peak observed at 1.15V must be discussed
3. From Figure 4b, although the E_{corr} of RuMoNi is more positive, however, the i_{corr} for this sample is much higher than Ni foam. So, how is it concluded that this electrode is a corrosion-resistant material?
4. The electrochemical tests should be discussed in more details for example the authors have to focus on EIS results and the changes in the parameters of the equivalent circuit used for fitting of these data
5. More corrosion studies are also necessary to conclude about the corrosion resistance of the samples and their changes over time.

Response to Reviewer #1

Comment.

Liu and co-workers report on a corrosion-resistant RuMoNi catalyst which is active for long-term seawater oxidation as well as seawater anion exchange membrane electrolyzer. First of all, seawater splitting especially when KOH is added, is tremendously more expensive than conventional water splitting, as such an electrolyzer requires a feed of KOH and seawater instead of only pure water, as the addition of only seawater will lead to increasing concentrations of all impurities. Therefore, the whole electrolyte must be replaced from time to time. If only pure water is used, the KOH can remain in the electrolyzer. Furthermore, seawater desalination (something that can always be done when seawater and electricity is available) adds basically no costs to the hydrogen production process. The authors have ignored all literature on this issue (see <https://doi.org/10.1039/D1EE00870F>, <https://doi.org/10.1039/D0EE03659E>, [10.1021/acssuschemeng.8b06779](https://doi.org/10.1021/acssuschemeng.8b06779), ...). This does not mean that seawater splitting must not be performed academically, but the claims of the introduction are rather overestimated, and it is not solving any real-world problem making it not useful for society. This must be stated in the introduction and critical reports must be cited. Additionally, many other issues need to be clarified before this manuscript is considered for a high-impact journal such as Nature communication.

Response. We thank the reviewer very much for the time in evaluating our work and the inspiring comments. We appreciate the reviewer's comment on the importance of seawater splitting for academic research. The reviewer is raising a very interesting point regarding whether seawater splitting can be a practical technology.

(1) Regarding economic value of seawater splitting. Actually, according to the recent literature discussing the economic value of seawater splitting and pure water splitting (e.g., *Energy Environ. Sci.* **14**, 4831-4839 (2021). *Energy Environ. Sci.* **14**, 3679-3685 (2021). *ACS Sustain. Chem. Eng.* **7**, 8006-8022 (2019). *Nat. Energy* **5**, 367-377 (2020). *Joule* **5**, 1921-1923 (2021)), the community has not reached a convincing or fixed conclusion about the prevailing technology up to now. As a result, both seawater splitting and water splitting are promising research directions and heavily studied by the community currently.

(2) Regarding addition of KOH. As for seawater splitting with KOH adding, although KOH will increase the cost, the alkaline environment created by KOH can keep the reaction running at a low potential to decrease the electricity cost. Note that in our work, we add KOH into the electrolyte before electrolysis only once. After that, we do not add KOH into the electrolyzer and only seawater (without KOH) is added into the electrolyte for subsequent seawater splitting.

Changes to the revised manuscript. We have taken your comments, added more

background information and cited some critical reports in the revised introduction. **On page 3**, “Seawater electrolysis leads to several promising research directions, such as desalinated, direct, and alkalized or acidified seawater electrolysis. It has been a heavily studied topic about the economic value of seawater electrolysis. For example, some analysts suggest that direct seawater electrolysis is not economic favorable (*Energy Environ. Sci.* **14**, 4831-4839 (2021). *Energy Environ. Sci.* **14**, 3679-3685 (2021). *ACS Sustain. Chem. Eng.* **7**, 8006-8022 (2019).), while some other studies suggest that direct seawater electrolysis shows low cost with economic benefit (*Nat. Energy* **5**, 367-377 (2020). *Joule* **5**, 1921-1923 (2021)). At this stage, the community has not reached a convincing or fixed conclusion about the preferable method, and more studies are needed on this topic, especially on the development of high-efficient and durable electrocatalysts for seawater electrolysis at high current density. Recently, Guo et al. designed a flow-type electrolyzer using abundant seawater resources (*Nat. Energy* **8**, 264-272 (2023)). In another work, Xie et al. reported the one-step hydrogen production from seawater (*Nature* **612**, 673-678 (2022)), indicating that the direct use of seawater in an industrial water electrolysis system, especially the anion exchange membrane (AEM) electrolyzer, is desirable.”

Comment 1. What is the oxidation state of Ru and Mo in RuMoNi precatalyst? Is it metallic? Also, the authors should clearly explain the bands related to the precatalysts and electrocatalysts in the Raman spectra.

Response 1. (1) Regarding oxidation state. The oxidation state of Ru and Mo in RuMoNi precatalyst can be seen in XPS spectra in Ru 3d (Figure R1) and Mo 3d (Figure R2), which show peaks at 279.5 and 227.8 eV, confirming the metallic state of Ru (*Chem. Lett.* **9**, 1537-1540 (1980)) and Mo (*J. Catal.* **36**, 11-22 (1975)). We also find doublets of Mo⁶⁺ and Mo⁴⁺ besides metallic Mo in RuMoNi precatalyst, suggesting the coexistence of molybdenum oxides. We provide Figure R1 and Figure R2 in the revised SI as Supplementary Fig. 5 and 6, and update the SI with related description.

(2) Regarding Raman bands. The band at 447 cm⁻¹ is the characteristic peak of Ni(OH)₂ (*J. Catal.* **36**, 11-22 (1975). *Nat. Catal.* **4**, 1050-1058 (2021)). The band at 730 cm⁻¹ is the anti-symmetric vibration of O-Mo-O (*J. Phys. Chem. B* **104**, 10059-10068 (2000)). Under the applied voltage of 1.3 V, 1.4 V, and 1.5 V, bands at 477 cm⁻¹ and 558 cm⁻¹ are the characteristic peaks of NiOOH (*Angew. Chem.* **132**, 8149-8154 (2020)). The detailed explanations have been updated in the revised SI.

Figure R1. The XPS spectrum of Ru $3d$ in the RuMoNi precatalyst. Ru shows a peak at 279.55 eV corresponding to the binding energy of $\text{Ru}^0 3d_{5/2}$, which indicates the existence of metallic Ru in the precatalyst. This figure is added as Figure S5 in the revised SI.

Figure R2. The XPS spectrum of Mo $3d$ in RuMoNi precatalyst. In this figure, we can find doublets of Mo^0 , Mo^{4+} , and Mo^{6+} with $3d_{5/2}$ bands centered at 227.84 eV, 229.25 eV, and 231.47 eV, respectively, which confirm the metallic state of Mo and suggest the coexistence of molybdenum oxides in the precatalyst. This figure is added as Figure S6 in the revised SI.

Figure R3. In-situ Raman spectra of the RuMoNi precatalyst at different applied potentials. The band at 447 cm^{-1} is the characteristic peak of Ni(OH)_2 . The band at 730 cm^{-1} is the anti-symmetric vibration of O-Mo-O. Under the applied voltage of 1.3 V, 1.4 V, and 1.5 V, bands at 477 cm^{-1} and 558 cm^{-1} are the characteristic peaks of NiOOH. This result indicates that the reconstructed area consists of NiOOH after electrochemical activation. This figure is added as Figure S10 in the revised SI.

Changes to the revised SI on page 14, “Ru shows a peak at 279.55 eV corresponding to the binding energy of $\text{Ru}^0 3d_{5/2}$, which indicates the existence of metallic Ru in the precatalyst”. “We can find doublets of Mo^0 , Mo^{4+} , and Mo^{6+} with $3d_{5/2}$ bands centered at 227.84 eV, 229.25 eV, and 231.47 eV, respectively, which confirm the metallic state of Mo and suggest the coexistence of molybdenum oxides in the precatalyst”. **On page 18,** “The band at 730 cm^{-1} is the anti-symmetric vibration of O-Mo-O. Under the applied voltage of 1.3 V, 1.4 V, and 1.5 V, bands at 477 cm^{-1} and 558 cm^{-1} are the characteristic peaks of NiOOH. This result indicates that the reconstructed area consists of NiOOH after electrochemical activation.”

Comment 2. In the main text, the authors state that Ru is doped in the precatalyst, which the reviewer presumes homogeneously distributed Ru in the structure (no related characterization is provided); however, in the electrocatalyst phase, RuO_2 can be found only at the surface. The reviewer wonders, if the only surface structure is reconstructed, what happens to the Ru that was beneath the top layer? Why can the authors only find Ni_4Mo in the inside region? It is also not clear what amorphous phase the authors find on the surface.

Response 2. (1) Regarding the Ru distribution and reconstruction. We apologize for insufficiently discussing the compositions of RuMoNi. As the reviewer mentioned,

the Ru is doped in the precatalyst. To identify the Ru distribution in the precatalyst, we test the XPS of Ru 3p before and after 2 min Ar⁺ etching. Before etching, the atomic content of Ru among metal elements is 1.37%, while Ru content decreases to below the detection limit after Ar⁺ etching. Figure R4 shows the increase of noises in XPS spectra of Ru 3p from exterior to interior, indicating that Ru mainly distributes on the surface of the precatalyst. During electrochemical oxidation, Ru on the surface reconstructs to RuO₂ which is one of the active phases in RuMoNi catalyst.

(2) Regarding Ni₄Mo from the inside region. After reconstruction, Ru on the surface reconstructs to RuO₂. NiOOH and NiMoO₄ form on the surface during the reconstruction process, while molybdenum oxides dissolve into the electrolyte. Thereafter, according to the HRTEM image, we find the Ni₄Mo mainly in the inside region.

(3) Regarding amorphous phase on the surface. For the amorphous phase on the surface, NiOOH shows the characteristic peak in Raman spectra (Supplementary Fig. 10), but no related signal can be found in XRD (Fig. 4e). As a result, the amorphous phase contains NiOOH. Amorphous NiOOH was also reported by other researchers in nickel-based catalysts (*Nat. Catal.* **2**, 763-772 (2019). *Adv. Mater.* **32**, 2001136 (2020)).

Figure R4. XPS spectra of Ru 3p. The top curve is from the surface of the sample while the bottom curve is from the interior of RuMoNi precatalyst after 2 min Ar⁺ etching.

Changes to the revised manuscript on page 6, “Furthermore, *in-situ* Raman spectroscopy shows peaks of NiOOH at 480 cm⁻¹ and 560 cm⁻¹ (Supplementary Fig. 10), but no related signal can be found in XRD (Fig. 4e). For the amorphous phase on the surface, the above results indicate that it contains NiOOH.”

Comment 3. The construction process of RuNiMo should be described in detail. How (MoO₄)²⁻ are stabilized on the surface during the reaction.

Response 3. We appreciate the reviewer's comments. **(1) Regarding detailed description of the reconstruction process of RuMoNi.** "According to XPS spectra of the RuMoNi precatalyst and electrocatalyst, during the reconstruction process, Ru⁰ is oxidized to Ru⁴⁺ (Supplementary Fig. 5, Supplementary Fig. 11). Ni⁰ on the surface is oxidized to Ni²⁺, while Ni⁰ exists in the interior region (Supplementary Fig. 7b, Supplementary Fig. 8b). Mo⁰, Mo⁴⁺ and Mo⁶⁺ exist in RuMoNi precatalyst, and Mo⁶⁺ in RuMoNi catalyst (Supplementary Fig. 6, Fig. 4d). Results from HRTEM (Fig. 1d-i), XRD (Fig. 4e), and Raman (Supplementary Fig. 10) reflect the composition changes during the reconstruction process. Ru on the surface reconstructs to RuO₂ active phases. Molybdenum oxides dissolve and form MoO₄²⁻ in the electrolyte. NiOOH and NiMoO₄ form on the surface. Ni₄Mo exists in the interior region." **We have added the above sentences on page 3 in the revised SI.**

(2) Regarding how MoO₄²⁻ anions are stabilized on the surface during the reaction. First, according to molecular dynamics simulation results from the literature (*Angew. Chem. Int. Ed.* **60**, 22740-22744 (2021)), the applied electric field drives the anions moving towards the anode and the polyanions such as sulfates and molybdates tend to be enriched above the electrode surface. There is also other literature reporting the electrostatic absorption of ions on the surface of active catalysts (*J. Am. Chem. Soc.* **144**, 3039-3049 (2022). *Angew. Chem. Int. Ed.* **134**, e202207279 (2022)). Therefore, an electrostatic force is one of the reasons that MoO₄²⁻ can be stabilized and absorbed on the catalyst surface. Second, as a constant atomic concentration of Mo in the electrolyte is detected by ICP-OES, the reversible dissolution and precipitation of MoO₄²⁻ and NiMoO₄ at the electrochemical interface are responsible for stabilizing the MoO₄²⁻.

We have updated the manuscript to reflect these points. On page 16, "Combined with the constant atomic concentration of Mo in the electrolyte detected by ICP-OES, the reversible dissolution and precipitation of MoO₄²⁻ and NiMoO₄ at the electrochemical interface is responsible for stabilizing the MoO₄²⁻. Under the electric field during electrolysis, the multivalent MoO₄²⁻ anions are preferentially absorbed on the anode through electrostatic force, and the enriched MoO₄²⁻ anions near the anode surface repels and blocks Cl⁻ by electrostatic repulsion."

Comment 4. Does the presence of nickel foam affect the electronic structure of the materials? Similarly, what is the Fe concentration in 1 M KOH? How much is Fe still present in the final electrocatalytic state? How does it influence electrocatalysis?

Response 4. (1) Regarding the influence of Ni foam on the electronic structure. Fig. 1a-b shows the nanorod morphology of the RuMoNi catalyst, in which the length of the

nanorod is more than 20 μm . There is a huge size difference between the nickel foam-RuMoNi interface (i.e., nanometer or smaller) and the RuMoNi catalyst (20 μm). Based on literature related to the interface between the substrate and the catalyst (*J. Am. Chem. Soc.* **133**, 7296-7299 (2011)), the presence of nickel foam has no influence on the electronic structure of the materials except the interfacial layer which only take a negligible percentage of the whole material.

(2) Regarding the concentration of Fe in the electrolyte and electrocatalyst, and its influence. We have taken your suggestion and tested the Fe concentration in 1.0 M KOH and 1.0 M KOH + seawater electrolyte (Table R1), which are 0.0975 ppm and 0.0675 ppm, respectively. As for Fe in the final electrocatalytic state, Fe cannot be detected through XPS and energy dispersive X-ray spectroscopy (Supplementary Fig. 4 and Fig. 1g), because the concentration is below the detection limit. Some researchers focused on the effect of Fe electrolyte impurities and provided evidence of the promotion effect of Fe (*J. Phys. Chem. C* **119**, 7243-7254 (2015). *J. Phys. Chem. C* **119**, 11475-11481 (2015)). In our study, RuMoNi shows much higher activity than MoNi in the same electrolyte (Figure R5, Table R2), indicating that the activity is dominated by Ru-based active sites, rather than Fe.

Figure R5. Polarization curves of RuMoNi and MoNi in a 1.0 M KOH + seawater electrolyte at a scan rate of 5 mV s^{-1} with 85% iR correction.

Table R1. Mass concentration of Fe impurities with standard deviation in 1.0 M KOH and 1.0 M KOH + seawater electrolyte.

Electrolytes	Mass concentration of Fe (ppm)	Standard deviation (ppm)
1.0 M KOH + seawater	0.0675	0.00829
1.0 M KOH	0.0975	0.00829

Table R2. Overpotentials of RuMoNi and MoNi catalyst at 100 mA cm⁻², 500 mA cm⁻², and 1,000 mA cm⁻².

Current density	RuMoNi (mV)	MoNi (mV)
100 mA cm ⁻²	291	405
500 mA cm ⁻²	397	540
1,000 mA cm ⁻²	484	670

Comment 5. What is the pH of the 1 M KOH (+ seawater) solution? How was it measured? Which values were used to convert it to RHE? A recent couple of reports in ACS Energy Letters state that the pH meter is not a good method to evaluate precise pH values.

Response 5. The pH values of 1.0 M KOH and 1.0 M KOH + seawater electrolyte are 14.13 and 14.10 respectively, tested by a pH meter. We have calculated the pH value based on the method reported by the literature in ACS energy letter (*ACS Energy Lett.* **6**, 3567-3571 (2021)), the pH value of 1.0 M KOH electrolyte at 20 °C is 14.06. If we take pH values of 14.13 or 14.10 (by the pH meter) and 14.06 to convert the voltage, the difference in potentials vs. RHE will be 2-4 mV, which is too small to affect the performance evaluation. Therefore, we use the pH value of 14.10 tested by a pH meter to convert the potential vs. Hg/HgO to potential vs. RHE.

We have updated the SI based on the suggestion. On page 5, “The pH values of 1.0 M KOH (14.13) and 1.0 M KOH + seawater (14.10) were tested by a pH meter, and this value was also calculated to be 14.06 according to a recent publication (*ACS Energy Lett.* **6**, 3567-3571 (2021)). In this work, we took the pH value of 14.10 to perform the potential conversion. Potentials were converted to potential vs. RHE after being measured vs. Hg/HgO by $E_{RHE} = E_{Hg/HgO} + 0.925$ V. The scan rate was 5 mV s⁻¹ for the cyclic voltammetry tests.”

Comment 6. As the authors show that the used seawater contains a mixture of several inorganic substances (Mg^{2+} , Ca^{2+} , Na^{2+} , sulfate) as well as they may be organic mixtures and microorganisms. Such a complex mixture present in the seawater can deeply affect the reconstruction process and performance or can be adsorbed in the active phase. How did the author remove/or avoid such a phenomenon? The authors do not even seem to observe any adsorption or interference from the salts. If they were precipitated out before, then it would be an extra cost to separate these products. Did they consider the separation step in the price calculation? Again, this is, in fact, not a significant advantage in comparison to direct water splitting using purified water.

Response 6. Thank the reviewer for raising this important point. **(1) Regarding the pre-processing of seawater.** Mg^{2+} and Ca^{2+} form precipitates after adding KOH into natural seawater, according to the low K_{sp} values of $Mg(OH)_2$ (1.5×10^{-11}) and $Ca(OH)_2$ (7.9×10^{-6}). After keeping still, the transparent electrolyte without precipitate, denoted as 1.0 M KOH + seawater electrolyte, was used to perform the electrochemical test. In the following electrochemical test in the 1.0 M KOH + seawater electrolyte, natural seawater without KOH or precipitation pre-process was added into the above electrolyte. Second, no additional procedure is used to get rid of other inorganic substances, organic mixtures, and microorganisms. We observed the reconstruction process and characterized the performance of electrocatalysts using the electrolyte prepared in this way. As shown by the electrochemical test, the catalyst is stable for such seawater electrolysis presumably due to the adsorbed MoO_4^{2-} anions on the catalyst, vigorous bubble releasing, and strong alkaline electrolyte.

(2) Regarding the price calculation. We agree with the reviewer that for the total price calculation, the separation cost should be considered. As what has been suggested by the reviewer and discussed in the Introduction and your general comment at the beginning, both seawater and purified water splitting are promising technologies that are under research currently. This study mainly focuses on the electricity cost to operate the electrolysis which is also used by U.S. DOE and other researchers in the field (*Nat. Nanotechnol.* **16**, 1371-1377 (2021). *Nat. Nanotechnol.* **14**, 1071-1074 (2019)). Using the same cost evaluation method, we didn't add the cost of the separation step to the price evaluation so that to make a fair comparison with others.

Changes in the revised SI. On page 5, "After adding KOH into natural seawater and settling down the precipitate, the transparent electrolyte was directly used as an electrolyte to perform electrochemical tests. In the following electrochemical test in the 1.0 M KOH + seawater electrolyte, natural seawater without KOH or precipitation was added into the electrolyte". **On page 6,** "During the durability test in 1.0 M KOH + seawater electrolyte, natural seawater without KOH or precipitation was pumped consistently into the electrolyte to simulate the real seawater electrolysis situation". **On page 10,** "Note 1. Calculations of AEM electrolyzer efficiency and H_2 cost. These

calculations only considered the electricity costs, based on the method proposed by literature.”

Comment 7. Tafel slopes must be measured via a steady-state method. How did the authors measure it? The description is missing in the manuscript.

Response 7. Thank you for the suggestion. We have used the steady-state responses for Tafel slope construction to replace the previous result derived from LSV curves. Both the results (Figure R6) and the description of the test have been updated in the revised manuscript.

Figure R6. Tafel slopes of the three electrocatalysts with chronoamperometry measurements. This figure is added as Figure 2e in the revised Manuscript.

Changes in the revised manuscript on page 11. Figure R6 replaced Fig. 2e driving from LSV curves. **Changes in the revised SI.** Related to Electrochemical measurements **on page 6.** “Tafel slope analyses were done with the steady-state response with a 100% iR drop compensation based on the overpotentials and current densities from 500 s chronoamperometry measurements at different applied voltages.”

Comment 8. Since (MoO₄)²⁻ was the phase that was repelling Cl⁻, did the authors use only RuMo or NiMo as references, especially NiMo, which would be more economically beneficial?

Response 8. Thank you for the instructive suggestions. We also regard the economically beneficial NiMo as a promising electrocatalyst, and have followed your suggestions and used NiMo to perform seawater electrolysis. During the chronopotentiometry (CP) test at a current density of 500 mA cm⁻² in 1.0 M KOH +

seawater electrolyte (Figure R7), NiMo can durably operate for more than 250 h, which confirms that MoO_4^{2-} is effective to repel Cl^- in NiMo, similar to the case of RuNiMo. However, analysis in Response 4 indicates that NiMo requires much higher overpotentials to achieve the same current densities than RuMoNi (e.g., 291 mV vs 405 mV to reach 100 mA cm^{-2}), resulting in an increased price of electricity cost in hydrogen production (Table R2, Figure R5). Because electricity cost is a major component in water electrolysis, RuMoNi with higher energy conversion efficiency from electricity to hydrogen is beneficial.

Figure R7. Durability test of the NiMo catalyst for over 250 h recorded at a current density of 500 mA cm^{-2} in a 1.0 M KOH + seawater electrolyte. This figure was added as Supplementary Fig. 25 in the revised SI.

Changes to the manuscripts on page 16, “ MoO_4^{2-} has also been demonstrated to have a protective effect in a Cl^- containing solution by reducing Cl^- adsorption and penetration of the corrosion-resistant layer, which was evidenced by the durability of NiMo catalyst at a current density of 500 mA cm^{-2} for more than 250 h in the CP test (Supplementary Fig. 25)”. **We also revised the SI on page 3.** “The NiMo electrocatalyst was synthesized by the same method as RuMoNi but without adding $\text{RuCl}_3 \cdot x\text{H}_2\text{O}$, and followed the same process of electrochemical activation.”

Comment 9. The reviewer missed the point of how the stability was performed. Was it a flow cell or did the authors add deionized water at a certain time interval during these tests?

Response 9. We thank the reviewer for the comments. Details of the stability tests have been added in the revised SI as follows.

In the part termed “Electrochemical measurements” on page 6, “Durability tests were conducted at a constant current density of 500 mA cm^{-2} for 0-3,000 h with a three-electrode electrochemical cell in a sequence of electrolytes at different temperatures. During the durability test in 1.0 M KOH + seawater electrolyte, natural seawater without KOH or precipitation was pumped consistently into the electrolyte to simulate the real seawater electrolysis situation. During the durability test in 1.0 M KOH + 2.0

M NaCl, 2.0 M NaCl solution was pumped into the electrolyte”. In the part termed “AEM electrolyzer fabrication” on page 7, “The durability test was carried out by CP technology at a current density of 500 mA cm⁻² for over 250 h, during which natural seawater without KOH or precipitation was directly added into the electrolyte.”

Comment 10. The authors should also look at the recent reports, e.g., <https://doi.org/10.1039/C8EE00927A>, where only in an alkaline solution, a current density of 1 A can be generated in much lower potentials.

Response 10. We thank the reviewer for mentioning this paper. This literature reported an active electrocatalyst (Ni, Fe) OOH, which achieves 1,000 mA cm⁻² at 1.657 V. This paper highlights the importance of high current density in water splitting which is consistent with our work. This paper and our study are using different electrolytes (alkaline water vs. seawater) and electrochemical cells (two-electrode electrochemical cell vs. AEM electrolyzer) and therefore it is not fair to compare the performance directly. As discussed in the beginning, both two technologies are promising for hydrogen production and deserve study.

Changes to the revised manuscript. We have added the reference (*Energy Environ. Sci.* **11**, 2858-2864 (2018)) on page 3. “Electrolysis at high current densities is crucial for practical applications, but the above problems become more serious.”

Comment 11. Depending on the 4-month stability period, the extrapolation of the data for 10 years seems to be mere speculation.

Response 11. This is an inspiring point. We agree with you that the extrapolation of the data for 10 years is speculation. We did this following research in other fields where similar extrapolation of the data are widely used, for example in fields of photovoltaics (*Nat. Commun.* **8**, 14068 (2017). *Nat. Commun.* **7**, 10808 (2016)) and electronic devices (*Nature Electronics* **3**, 466-472 (2020). *Nat. Commun.* **11**, 1439 (2020), *Nat. Commun.* **12**, 5198 (2021)). To some extent, extrapolation can predict the retention characteristic of materials and devices based on the experimental results. Therefore, we humbly request to keep this sentence and bring this method to inspire colleagues in the electrolysis community.

We have updated the manuscript on page 13 to make it clear. “It is speculated that after 10-year operation the voltage would increase by only ~56 mV (Supplementary Fig. 24), which suggests the stability of RuMoNi to a certain extent.”

Comment 12. The Cdl was measured in KOH solution instead of KOH+seawater. Why? Does this influence the ECSA of the catalyst? The authors in the ESI state that “the ideal planar electrode has a Cdl of 0.04 mF cm⁻²” while they used a much more porous 3D nickel foam for the experiments.

Response 12. We have tested the electrochemical surface area (ECSA) in both 1.0 M KOH and 1.0 M KOH + seawater electrolyte (Figure R8), where the ECSA showed a negligible difference. Based on the fact that the double layer capacitance of an ideal planar electrode, C_{dl}=0.04 mF cm⁻², is corresponding to 1.0 M KOH electrolyte, we measured the C_{dl} of RuMoNi in KOH solution instead of KOH + seawater. We are sorry that the symbols and description of the R_f calculation equation may be misunderstood. We use the C_{dl} of an ideal planar electrode (defined as C_s) to normalize the C_{dl} of the porous 3D nickel foam in experiments.

Figure R8. CV curves of RuMoNi in a) 1.0 M KOH, b) 1.0 M KOH + seawater. c) Capacitive currents at 0.87 V vs. RHE against scan rates for RuMoNi in 1.0 M KOH and 1.0 M KOH + seawater. This figure was added as Supplementary Fig. 13 in the revised SI.

Changes in the revised SI on page 6, “The double layer capacitance (C_{dl}) was calculated by CV measurements at scan rates from 10 to 50 mV s⁻¹ in the non-faradaic region in a 1.0 M KOH electrolyte, because of the negligible difference in C_{dl} of RuMoNi electrode in 1.0 M KOH and 1.0 M KOH + seawater (Supplementary Fig. 13). The electrochemical active surface areas (ECSA) were obtained based on C_{dl}. The C_{dl} is estimated using the equation: $C_{dl} = \frac{\Delta j}{\Delta v} = \frac{j_a - j_c}{2 \cdot v}$, where j_a and j_c are the anodic and cathodic current densities, respectively, recorded at a potential of 1.125 V vs. RHE, and v is the scan rate (Supplementary Fig. 11). An ideal planar electrode has a C_{dl} of 0.04 mF cm⁻², defined as C_s = 0.04 mF cm⁻². The roughness factor (R_f) can be calculated using the equation: $R_f = \frac{C_{dl}}{C_s}$. The specific current density (j_{specific}) was calculated by the following equation: $j_{specific} = \frac{j}{R_f}$.”

Response to Reviewer #2

Comment.

The idea is worthy of research but it needs to be done more detailed. I propose these suggestions:

Response. We thank the reviewer very much for the effort in evaluating our work and the helpful comments. We appreciate the reviewer writing that “the idea is worthy of research”. We have provided more details regarding experiments as requested, as shown in the following responses.

Comment 1. There are lots of papers reported about the use of different chloride barrier layers to increase the selectivity of anodes over OER. However, in introduction section we do not see a concise bibliography of these works.

Response 1. We have taken this suggestion and added discussions on the use of chloride barrier layers in the introduction section referring to the following literature (*ACS Catal.* **11**, 1316-1330 (2021). *Langmuir* **36**, 5227-5235 (2020). *J. Am. Chem. Soc.* **140**, 10270-10281 (2018). *Adv. Mater.* **33**, 2101425 (2021)).

Changes to the manuscript on page 4, “Beside durability, how to increase the selectivity of anode over OER is another critical issue in seawater electrolysis. In this regard, chloride barriers are widely used, and some electrode selectivity to O₂ production was enhanced to ~100%^{23,30,31}. For example, SiO₂ overlayer has been introduced as an effective barrier that blocks the transport of Cl⁻ and increases the selectivity of the desired OER²². Ma et al.³² have studied the effect of a sulfate additive on stable alkaline seawater oxidation and found that sulfate anions are preferentially absorbed on the anode surface to repel Cl⁻ and achieve high selectivity. Although those strategies show significant selectivity improvement, the corrosion of the conductive substrate in the saline electrolyte is still challenging. For example, sulfate anions have been shown to accelerate the corrosion of the electrocatalyst substrate because metal sulfates are unstable products that would be further oxidized to hydroxides or chlorides, and sulfate anions are released again and restart another cycle, finally resulting in the degradation of the electrode^{33,34}.”

Comment 2. In figure 2a, the peak observed at 1.15V must be discussed.

Response 2. The linear sweep voltammetry (LSV) curves in Figure 2a were derived from the backward scan of the cyclic voltammetry (CV). We think the reviewer might be referring to the peak near 1.3 V which corresponds to the reduction peak of Ni-sites (*Energy Environ. Sci.* **13**, 3439-3446 (2020). *Electrochim. Acta* **51**, 3609-3621 (2006)). Interpretation of this peak has been added to the caption of Figure 2a.

Changes to the manuscript on page 11, “Figure 2. OER performance in alkaline seawater at high current densities. a) Polarization curves of RuMoNi, RuO₂, and Ni Foam in a 1.0 M KOH + seawater electrolyte at a scan rate of 5 mV s⁻¹ with 85% *iR* correction. The peak of RuMoNi near 1.3 V corresponds to the reduction peak of Ni-sites.”

Comment 3. From Figure 4b, although the E_{corr} of RuMoNi is more positive, however, the i_{corr} for this sample is much higher than Ni foam. So, how is it concluded that this electrode is a corrosion-resistant material?

Response 3. The corrosion potential (E_{corr}) is a critical factor that reflects the dynamics of the corrosion process. The corrosion current (I_{corr}) indicates the reaction rate of corrosion. I_{corr} can be affected by many factors like geometric area, roughness, specific area of the electrode, and others. As a comparison, E_{corr} is a more intrinsic indicator than I_{corr} . In our work, we have standardized the Tafel plot test of Ni foam and RuMoNi electrocatalyst by using electrodes with the same geometric area and deriving the specific current density by ECSA normalizing. After such standardization, as shown in the updated Tafel plot in Figure R9, the $j_{\text{corr,specific}}$ of RuMoNi is lower than Ni foam. It concludes that RuMoNi electrode is a corrosion-resistant material.

Figure R9. Tafel plots of RuMoNi and Ni Foam in a 1.0 M KOH + 0.5 M NaCl electrolyte. The j_{specific} is the current density normalized by ECSA. This figure was added as Figure 4b in the revised manuscript.

Changes to the revised SI on page 6, “The corrosion behaviors of different electrodes were studied in the three-electrode electrochemical cell using RuMoNi and Ni foam electrodes with a geometric area of 1 cm × 1 cm as the working electrodes. The polarization tests were performed in 1.0 M KOH + 0.5 M NaCl electrolyte.”

Comment 4. The electrochemical tests should be discussed in more details for example the authors have to focus on EIS results and the changes in the parameters of the equivalent circuit used for fitting of these data.

Response 4. We thank the reviewer for this insightful comment. We have analyzed the electrochemical test results, especially the EIS. The smallest charge transfer resistance of RuMoNi electrocatalysts is confirmed by changes in the parameters of the equivalent circuit used for data fitting (Figures R10, R11, Table R3). Discussion of other electrochemical analyses, such as Tafel slopes, Tafel plots, and corrosion rates has also been updated. The revision related to EIS was as follows.

Figure R10. Enlarged electrochemical impedance spectra of RuMoNi, RuO₂, and Ni Foam. This figure was added as Supplementary Fig. 16 in the revised SI.

Figure R11. The equivalent circuit for impedance spectra data of three catalysts (Note: the fitting software is Z-view). R_s is the resistance of the solution, R_c is the resistance of the corrosion layer and the R_{ct} is the charge transfer resistance on the electrode surface. CPE1 and CPE2 stand for the constant phase elements. This figure was added as Supplementary Fig. 17 in the revised SI.

Figure R12. Electrochemical impedance spectra of RuMoNi, RuO₂, and Ni Foam. Figure 2d was replaced by Figure R12 in the revised manuscript.

Table R3. Fitting data of EIS data of RuMoNi, RuO₂, and Ni Foam. This table was added as Table S4 in the revised SI.

Catalyst	R _s (Ω)	R _c (Ω)	R _{ct} (Ω)
RuMoNi	0.31	0.15	1.60
RuO ₂	0.28	0.12	5.54
Ni Foam	0.21	0.04	24.72

R_s is the resistance of the solution. R_c is the resistance of the corrosion layer and R_{ct} is the charge transfer resistance on the electrode surface. The charge transfer resistance of RuMoNi is much smaller than those of RuO₂ and Ni Foam.

Changes to the revised manuscript. Figure 2d was replaced by Figure R12 with both raw data and fitting data. **On page 10**, “The electrochemical impedance spectroscopy (EIS) results demonstrate the efficient charge transfer of the RuMoNi electrocatalyst in 1.0 M KOH + seawater electrolyte (Fig. 2d), whose charge transfer resistance is much smaller than those of the benchmark RuO₂ and Ni Foam based on the equivalent circuit in Supplementary Fig. 16 and fitting data in Table S4”. We added Figures R10, R11 and Table R3 to the revised SI.

Comment 5. More corrosion studies are also necessary to conclude about the corrosion resistance of the samples and their changes over time.

Response 5. We thank the reviewer for the kind suggestion and have performed more corrosion studies per your suggestion. To compare the difference in corrosion resistance of the samples, we use the dissolution rate of Ni from the electrode to the electrolyte to quantify the corrosion rate. From Table R4, the corrosion rate of Ni foam is 16.9 times higher than the RuMoNi electrode in the first hour, and rises to 463.5 times higher than the RuMoNi in the second hour. Therefore, the corrosion resistance of RuMoNi is higher than Ni. And during the stability test, the Ni foam electrode shows an increase in corrosion rate and cannot persist for more than 10 h (Supplementary Fig. 18). In contrast, RuMoNi electrode no longer shows any Ni dissolution into electrolyte according to the testing results in 20 h and 100 h. Figure R13 shows the good consistency of open circuit potential (OCP) of RuMoNi before and after 10 h stability test in 1.0 M KOH + seawater at a current density of 500 mA cm⁻², which also demonstrates the good resistance of RuMoNi electrode with negligible degradation over time.

Table R4. Dissolution rate of Ni from electrode to electrolyte during stability test at 500 mA cm⁻² in 1.0 M KOH + seawater electrolyte. This table was added as Table S5 in revised SI.

Time	Ni	RuMoNi	Ratio
(h)	(mg/h)	(mg/h)	Ni to RuMoNi
1	2.7057	0.1632	16.9
2	16.1197	0.0347	464.5
20	-	~0	-
100	-	~0	-

Figure R13. Open circuit potential (OCP) of RuMoNi before and after 10 h stability test in 1.0 M KOH + seawater at a current density of 500 mA cm⁻². This figure was added as Supplementary Fig. 20 in the revised SI.

Changes to the revised manuscript on page 12, “By testing the corrosion rate of electrodes and changes of open circuit potential (OCP) along the CP test, RuMoNi with high corrosion resistance shows a negligible degradation over time (Table S5, Supplementary Fig. 20).” **On page 6 in the revised SI,** “The open circuit potential (OCP) was measured after the electrode exposed in the electrolytes for several hours.”

REVIEWERS' COMMENTS

Reviewer #1 (Remarks to the Author):

The reviewer is glad that the authors are aware of controversies related to seawater splitting and appreciate them for stating that the focus of their work is to contribute academically towards developing durable electrocatalysts for seawater electrolysis. The authors have revised the manuscript significantly, and most of the critical issues have now been resolved. Thus, this manuscript can be accepted for publication.

A short remark: In the SI, the authors have used the value of a planar electrode for the normalization of Cdl measurements (0.04 mF cm^{-2}). However, this value only holds suitable for electrodes such as FTO or ITO, not for the 3D porous Ni foams (see, J. Am. Chem. Soc. 2013, 135, 16977–16987).

Response to Reviewer #1

Reviewer #1 (Remarks to the Author):

The reviewer is glad that the authors are aware of controversies related to seawater splitting and appreciate them for stating that the focus of their work is to contribute academically towards developing durable electrocatalysts for seawater electrolysis. The authors have revised the manuscript significantly, and most of the critical issues have now been resolved. Thus, this manuscript can be accepted for publication.

Response: We thank the reviewer very much for your recommendation.

A short remark: In the SI, the authors have used the value of a planar electrode for the normalization of C_{dl} measurements (0.04 mF cm^{-2}). However, this value only holds suitable for electrodes such as FTO or ITO, not for the 3D porous Ni foams (see, J. Am. Chem. Soc. 2013, 135, 16977–16987).

Response: We thank the reviewer for your nice suggestion.

Regarding to the normalization of C_{dl} measurements, $C_s = 0.04 \text{ mF cm}^{-2}$ is indeed the value of a planar electrode based on the typical reported value of Ni-based electrode (J. Am. Chem. Soc. 2013, 135, 16977–16987). And using voltammetry to get the ECSA of the porous materials, such as Ni foams electrode may generate errors to some extent, while it is still an effective approach for the validity of the internal comparison (I. Electroanal. Chem., 321 (1992) 353-376). The followings are some references using voltammetry to study the ECSA of porous electrode (Energy Environ. Sci. 15, 4647-4658 (2022), Nat. Commun. 10, 5106 (2019)). Additionally, the linear relationship between current density and scan rate from experiments is consistent with the description of the Helmholtz model for the electric double layer capacitance, which is the principle of voltammetry approach. Based on your suggestion, we have provided the Roughness Factor in Supplementary Table 2 and added the fitted correlation coefficients (close to 1) in the captions of Figs. R1 and R2.

Changes to the revised SI. On page 10, “Supplementary Figure 13. CV curves of RuMoNi in a) 1.0 M KOH, b) 1.0 M KOH + seawater at $20 \pm 2 \text{ }^\circ\text{C}$. c) Capacitive currents at 0.87 V vs. RHE against scan rates for RuMoNi in 1.0 M KOH and 1.0 M KOH + seawater. The R^2 values for the lines of RuMoNi in 1.0 M KOH and 1.0 M KOH + seawater are 0.999, and 0.992, respectively”. **On page 11**, “Supplementary Figure 14. CV curves of

a) RuO₂, b) Ni foam, and c) RuMoNi. d) Capacitive currents at 1.125 V vs. RHE against scan rate for RuMoNi, RuO₂, and Ni foam at 20 ± 2 °C. The R² values for the lines of RuMoNi, RuO₂ and Ni foam are 0.992, 0.998, and 0.999, respectively”.

Figure R1. CV curves of RuMoNi in a) 1.0 M KOH, b) 1.0 M KOH + seawater at 20 ± 2 °C. c) Capacitive currents at 0.87 V vs. RHE against scan rates for RuMoNi in 1.0 M KOH and 1.0 M KOH + seawater. The R² values for the lines of RuMoNi in 1.0 M KOH and 1.0 M KOH + seawater are 0.999, and 0.992, respectively.

Figure R2. CV curves of a) RuO₂, b) Ni foam, and c) RuMoNi. d) Capacitive currents at 1.125 V vs. RHE against scan rate for RuMoNi, RuO₂, and Ni foam at 20 ± 2 °C. The R² values for the lines of RuMoNi, RuO₂ and Ni foam are 0.992, 0.998, and 0.999, respectively.